# Structural basis for the assembly and energy transfer between the cyanobacterial PSI core and the double-layered IsiA proteins

Long Si[1,2], Yingyue Zhang[3], Xiaodong Su[1,2], Xuelin Zhao[1], Xiaomin An[1], Lu-Ning Liu [3,4] ✉, Peng Cao [1,5,6] ✉ & Mei Li [1] ✉

Iron-limitation is a common stress factor in natural environments. To survive under iron-starved conditions, cyanobacteria overexpress iron stress-induced protein A (IsiA), which is crucial for light-harvesting and photoprotection. Multiple IsiA proteins form a single- or double-layered architecture encircling the photosystem I (PSI) core, forming various PSI-IsiA supercomplexes. The assembly and energy transfer mechanisms of double-layered PSI-IsiA supercomplexes remain unelucidated. Here, we present high-resolution structures of two PSI-IsiA supercomplexes isolated from the cyanobacterium *Thermosynechococcus elongatus* BP-1 cultured under iron-starved conditions. The $PSI_3$-$IsiA_{43}$ complex contains a trimeric PSI core surrounded by 43 IsiA subunits assembled into a closed double-ring. The $PSI_1$-$IsiA_{13}$ complex contains 13 IsiA proteins arranged in a double-layered architecture attached to the monomeric PSI core. Atomic force microscopy demonstrates the presence and distribution of different PSI-IsiA complexes within native thylakoid membranes isolated from iron-starved cells. Our findings provide insights into the structural variability and adaptive mechanisms of PSI-IsiA complexes.

Cyanobacteria are among the earliest life forms on Earth, emerging approximately 2.4 billion years ago, and possess the ability to produce oxygen through oxygenic photosynthesis. Two essential components of this process are photosystems I and II (PSI and PSII), both of which are large membrane-embedded pigment-protein complexes, with their core complexes driving light-driven charge separation and electron transfer[1–4]. These core complexes usually collaborate with their peripheral antennae, which contain a high concentration of pigments and are capable of absorbing photons across different wavelengths of light, thereby significantly enhancing the absorption cross-section of the photosynthetic units[5]. Photosynthetic organisms adapted to different living environments have developed diverse peripheral antennae to efficiently capture solar energy[6–19]. While most plants and algae employ membrane-embedded pigment-protein complexes associated with PSI and PSII cores as peripheral antennae, cyanobacteria utilize distinct antennae called phycobilisomes (PBS), which are large water-soluble pigment-protein complexes. These PBS are primarily located on the cytoplasmic side of PSII, where they harvest and transfer photon energy to the PSII core[20,21]. In addition, certain cyanobacterial species possess membrane-embedded pigment-protein complexes, such as iron stress-induced protein A (IsiA), which functions as PSI antenna under stress conditions[22–24].

Iron availability is one of the main factors limiting the primary production of aquatic photosynthetic organisms, and iron limitation is a common stress constantly encountered by cyanobacteria living in natural aquatic environments[25]. In response to iron stress,

[1]State Key Laboratory of Biomacromolecules, Institute of Biophysics, Chinese Academy of Sciences, Beijing, China. [2]University of Chinese Academy of Sciences, Beijing, China. [3]Institute of Systems, Molecular and Integrative Biology, University of Liverpool, Liverpool, UK. [4]MOE Key Laboratory of Evolution and Marine Biodiversity, Frontiers Science Center for Deep Ocean Multispheres and Earth System & College of Marine Life Sciences, Ocean University of China, Qingdao, China. [5]College of Chemistry and Life Science, Beijing University of Technology, Beijing, China. [6]Institute of Matter Science, Beijing University of Technology, Beijing, China. ✉e-mail: luning.liu@liverpool.ac.uk; pengcao@bjut.edu.cn; meili@ibp.ac.cn

cyanobacteria induce the expression of IsiA and IsiB (flavodoxin, Fld) proteins to cope with stressful environments and ensure their normal growth[26–29]. Flavodoxin, which binds to a flavin mononucleotide molecule, acts as a functional substitute for the PSI electron acceptor ferredoxin, compensating for the loss of ferredoxin due to iron deficiency[30]. IsiA is homologous to the PSII core subunit CP43, which shares a similar architecture with six transmembrane helices; it is thus also termed CP43'. IsiA subunits can assemble into single-, double-, or multi-layered structures around the PSI core in vivo[31]. IsiA binds 17 chlorophyll (Chl) molecules in total, and the spectroscopic properties of PSI-IsiA supercomplexes have demonstrated that IsiAs can efficiently absorb light energy and transfer it to the PSI core[23,31–33]. These findings shed light on the intricate dynamics of light absorption and transfer within these complexes[34–37]. In addition, IsiA proteins alone can self-assemble into IsiA oligomers, which have been suggested to play a role in energy quenching[31,33,38,39].

A few cryo-electron microscopy (cryo-EM) structures of various cyanobacterial PSI-IsiA complexes have been reported previously, including single-ring PSI$_3$-IsiA$_{18}$ complexes from the mesophilic cyanobacteria *Synechocystis* sp. PCC 6803[40] and *Synechococcus elongatus* PCC 7942[41], as well as from the thermophilic cyanobacterium *Thermosynechococcus vulcanus* (*T. vulcanus*)[42]. In addition, the interaction of Fld with the PSI-IsiA supercomplex has recently been elucidated through the *Synechococcus* PSI-IsiA-Fld cryo-EM structure, revealing that three Fld proteins bind symmetrically to the trimeric PSI core[41]. Recently, structural analysis of the PSI$_1$-IsiA$_6$ complex from *Anabaena* sp. PCC 7120 showed that a monomeric PSI core lacking the PsaL subunit binds to six IsiA proteins, with the C-terminal domain of one IsiA (IsiA2) folding in a manner similar to that of PsaL and occupying its position[43]. Although substantial structural information is available for single-ring PSI-IsiA complexes, the nature of the double-ring PSI-IsiA assembly has not been characterized, and the precise mechanisms underlying the formation, function, and variability of cyanobacterial photosynthesis under iron-limiting stress remain unclear.

Here, we determine the structural organization and assembly of double-layered PSI-IsiA complexes using single-particle cryo-EM, atomic force microscopy (AFM), and complementary biochemical analyses. Our findings provide key insights into the structural and functional dynamics of cyanobacterial PSI-IsiA supercomplexes adapted to varying levels of iron stress.

## Results

### Iron-stress cell culture and sample characterization
The PSI-IsiA supercomplexes were purified from the thermophilic cyanobacterium *Thermosynechococcus elongatus* BP-1 (*T. elongatus*). To induce iron deficiency, we first cultured *T. elongatus* cells in the BG-11 medium containing standard $Fe^{2+}$ concentration (0.021 mM) and then transferred the cells (Fe$^+$ cells) into iron-deficient medium at two different dilution ratios (1/25 and 1/60, v/v). The different treatments resulted in two iron-limited culture conditions (moderate and severe), producing two types of cells, Fe$^-$ cell-1 and Fe$^-$ cell-2, respectively. Moreover, we further cultured Fe$^-$ cell-2 in iron-supplemented medium to obtain Fe$^{-/+}$ cells. Our electrophoresis analysis of these cells demonstrated that IsiA accumulates in both Fe$^-$ cells, and its protein level increases with iron concentration decreasing (Supplementary Fig. 1a). In addition, we measured the absorption spectra of these cells (Supplementary Fig. 1b), and found that both Fe$^-$ cell-1 and Fe$^-$ cell-2 exhibit a blue shift in the Q$_y$ absorption band of chlorophyll *a*, with Fe$^-$ cell-2 exhibiting a more pronounced shift than Fe$^-$ cell-1 of approximately 7 nm compared to Fe$^+$ cells. In contrast, Fe$^{-/+}$ cells show a Q$_y$ absorption band of chlorophyll *a* at 681.5 nm and a greatly reduced IsiA accumulation, similar to Fe$^+$ cells (Supplementary Fig. 1). These observations suggest that the IsiA protein level is directly related to the iron availability, and the Q$_y$ absorption peak in cyanobacterial cells could be used as an indicator of increased IsiA expression, consistent

with the earlier reports linking IsiA accumulation to changes in spectral properties[31,32,44].

We then fractionated the solubilized thylakoid membranes from three cell types (Fe$^-$ cell-1, Fe$^-$ cell-2 and Fe$^{-/+}$ cell) using sucrose density gradient centrifugation (Supplementary Fig. 2a) and characterized the major PSI fractions through negative staining electron microscopy. We found that the Fe$^-$ cell-1 culture conditions predominantly resulted in single-ring PSI-IsiA complexes, namely PSI$_3$-IsiA$_{18}$ (Supplementary Fig. 2b), whereas the Fe$^-$ cell-2 cultivation led to the formation of double-ring PSI-IsiA complexes, namely trimeric PSI$_3$-IsiA$_{43}$ and monomeric PSI$_1$-IsiA$_{13}$ (Supplementary Fig. 2c, d). In comparison, Fe$^{-/+}$ cells primarily form the trimeric PSI core (PSI$_3$) (Supplementary Fig. 2e), likely due to the reduction of IsiA proteins (Supplementary Fig. 1a). These findings suggest that the formation of either a single or a double ring of IsiA is primarily determined by the extent of iron limitation, and cyanobacterial PSI and PSI-IsiA complexes exhibit structural reversibility in response to the changes in iron concentration.

Next, we analyzed the protein composition and spectral properties of these PSI-IsiA complexes (Supplementary Fig. 3). Room-temperature absorption spectra showed that both the PSI$_3$-IsiA$_{18}$ and PSI$_3$-IsiA$_{43}$ complexes exhibited a Q$_y$ absorption band of chlorophyll *a* at 673 nm, whereas the PSI$_3$ sample showed a peak at 679 nm (Supplementary Fig. 3b), similar to the spectral profiles of the Fe$^-$ and Fe$^+$ cells (Supplementary Fig. 1b). Fluorescence emission kinetics measured via P700 oxidation demonstrated that the PSI$_3$, PSI$_3$-IsiA$_{18}$, and PSI$_3$-IsiA$_{43}$ complexes possess increased P700 oxidation rates, implying that these PSI complexes exhibit increased light absorption cross-section (Supplementary Fig. 3c), in line with a previous report[32]. HPLC analysis of both the PSI$_3$-IsiA$_{43}$ and PSI$_1$-IsiA$_{13}$ samples identified three major pigment peaks belonging to zeaxanthin (Zea), chlorophyll *a* (Chl *a*), and *β*-carotene (BCR) (Supplementary Fig. 3d), similar to the previously reported pigment composition of PSI$_3$-IsiA$_{18}$[41]. Together, these findings suggest that multiple types of PSI-IsiA supercomplexes were formed in *T. elongatus* under different iron-deficient conditions, and that these IsiA proteins function as peripheral antennae of the PSI core with high efficiency in light-harvesting and energy transfer.

### Architecture of PSI$_3$-IsiA$_{43}$ and PSI$_1$-IsiA$_{13}$ complexes
To gain further insights into the structural assembly of these double-layered PSI-IsiA complexes, we solved the cryo-EM structures of PSI$_3$-IsiA$_{43}$ and PSI$_1$-IsiA$_{13}$ at resolutions of 3.4 Å and 3.5 Å, respectively (Supplementary Figs. 4–6 and Supplementary Table 1). Our structural analysis showed that the PSI$_3$-IsiA$_{43}$ complex comprises a trimeric PSI core surrounded by a double-layered IsiA ring with overall dimensions of approximately $380 \times 380 \times 110$ Å (Fig. 1a–d). The inner IsiA ring consists of 18 IsiA subunits arranged in the same three-fold symmetry as the trimeric core. We designated the IsiA subunit on the PsaK pole of one PSI core (Core-1, Fig. 1c) as IsiA-i-1 and the 18 inner IsiA proteins as IsiA-i-1 to IsiA-i-18 in an anticlockwise direction when viewed from the lumenal side. The architecture of this part closely resembles the previously reported structures of PSI$_3$-IsiA$_{18}$[40–42], with the IsiA ring exhibiting a slight rotational shift (Supplementary Fig. 7a). The outer ring consists of 25 IsiA subunits that asymmetrically bind to the outer face of the inner IsiA ring, designated IsiA-o-1 to IsiA-o-25, starting from the IsiA-i-1 pole. Consistent with previously reported IsiA structures[40–42], each IsiA protomer possesses 6 transmembrane helices and a short C-terminal helix located on the cytoplasmic side and coordinates 17 Chl *a* and 4 carotenoid molecules. All 43 IsiA subunits share a similar orientation, with their short C-terminal helices extending toward the PSI core. This structural observation suggests that the specific orientation of IsiA subunits is critical for the well-ordered assembly of the entire complex.

The PSI$_1$-IsiA$_{13}$ structure consists of a monomeric core that lacks PsaL and binds 13 IsiA subunits arranged in two layers attached to the PSI core, with 6 copies in the inner layer and 7 copies in the outer layer

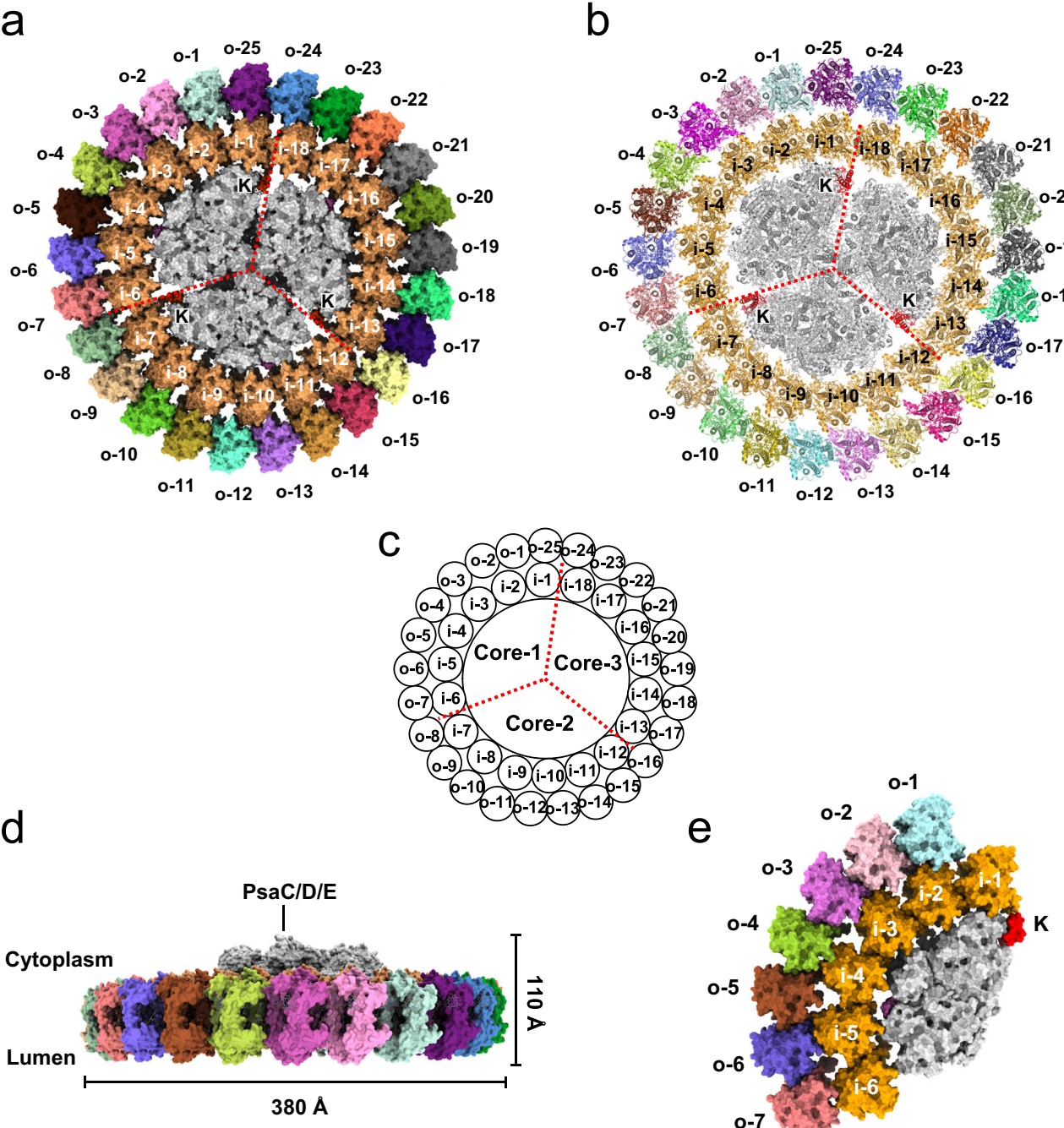

**Fig. 1 | Overall structures of PSI₃-IsiA₄₃ and PSI₁-IsiA₁₃.** **a** The PSI₃-IsiA₄₃ complex shown in surface mode and viewed from the lumenal side. Red dashed lines separate the three symmetrical parts, each containing one PSI core (shown in gray) and six inner IsiAs (shown in orange). The IsiA subunits in the inner ring are labeled as i-1-i-18. The IsiA proteins in the outer ring are shown in different colors and labeled as o-1-o-25. The three PsaK subunits are labeled with the letter "K". **b** Cartoon presentation of the PSI₃-IsiA₄₃ complex viewed from the lumenal side, using the same color codes as in (**a**). The PsaK subunits are labeled and colored in red. **c** Cartoon illustration of the PSI₃-IsiA₄₃ complex showing the subunit numbering. The three PSI cores are labeled as Core-1, Core-2, Core-3. **d** Side view of the PSI₃-IsiA₄₃ complex along the membrane plane. The cytoplasmic extrinsic subunits (PsaC, PsaD and PsaE) are indicated. **e** Lumenal side view of the PSI₁-IsiA₁₃ complex. PsbK is colored in red and labeled. IsiA subunits in the inner and outer layers are labeled as i-1-i-6 and o-1-o-7.

(Fig. 1e). We also observed a density blob adjacent to the IsiA-o-7 protein, which presumably corresponds to another IsiA subunit, indicating that an additional IsiA subunit (IsiA-o-8) is also present in the PSI₁-IsiA₁₃ complex (marked by the red arrow in Supplementary Fig. 5b). However, the poor density suggests that this IsiA protein appears to be highly mobile, and thus, was not modeled into our structure. When superimposed with each of the three PSI cores in the PSI₃-IsiA₄₃ structure, PSI₁-IsiA₁₃ aligned well with only one PSI moiety

(designated as PSI-moiety-1; Fig. 2a) containing Core-1, IsiA-i-1 to IsiA-i-6, and IsiA-o-1 to IsiA-o-7. In comparison, superimposition with the other two PSI cores showed that the outer IsiA subunits in the PSI₁-IsiA₁₃ structure exhibit a noticeable rotational shift of approximately 5° compared to the corresponding IsiA subunits of PSI₃-IsiA₄₃ (Fig. 2a and Supplementary Fig. 7b).

We next compared the PSI₁-IsiA₁₃ and the PSI-moiety-1 of the PSI₃-IsiA₄₃ complex to analyze their structural and compositional

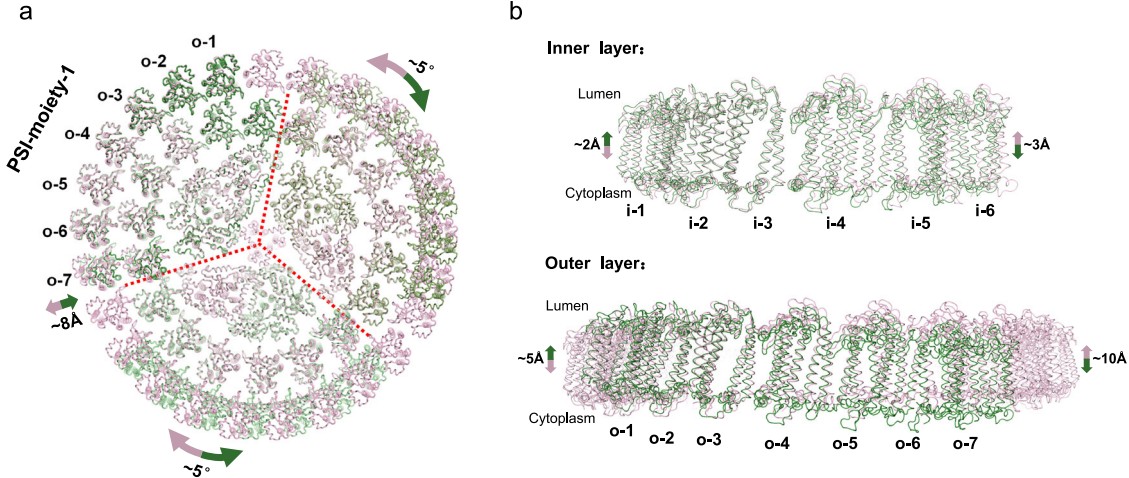

**Fig. 2 | Structural comparison of PSI$_3$-IsiA$_{43}$ and PSI$_1$-IsiA$_{13}$. a** The PSI$_1$-IsiA$_{13}$ structure (colored in green) was superimposed onto the three symmetrically arranged PSI core parts of PSI$_3$-IsiA$_{43}$ (colored in pink), viewed from the lumenal side. **b** Superposition of the inner IsiAs (IsiA-i-1-IsiA-i-6) and the outer IsiAs (IsiA-o-

1-IsiA-o-7) from PSI$_3$-IsiA$_{43}$ and PSI$_1$-IsiA$_{13}$, viewed from the membrane plane. Alignment was based on PsaA subunits. Shifts of IsiA proteins are indicated by arrows, and the shifting distances and rotation angles are labeled.

differences. We found that they exhibit similar subunit and cofactor compositions, except that the PsaL subunit is absent in the PSI$_1$-IsiA$_{13}$ complex, which was likely lost during our purification procedure due to its weak interactions with other core subunits. Despite these similarities, the structures of PSI$_1$-IsiA$_{13}$ and PSI-moiety-1 show notable differences (Fig. 2a and Supplementary Fig. 8). Compared to PSI$_3$-IsiA$_{43}$, the subunits IsiA-i-5 ~ 6 and IsiA-o-5 ~ 7 of PSI$_1$-IsiA$_{13}$ exhibited a noticeable shift (~8 Å) within the membrane toward the core, potentially hindering the incorporation of additional IsiA proteins. As a result, PSI$_1$-IsiA$_{13}$ may be unable to form an intact PSI$_3$-IsiA$_{43}$ complex (Fig. 2a). However, it cannot be ruled out that the trimerization reshapes the monomeric conformation, ensuring the formation of complete IsiA rings. In contrast to the relatively flat transmembrane region of PSI$_3$-IsiA$_{43}$, the membrane-spanning region of PSI$_1$-IsiA$_{13}$ exhibited a more pronounced curvature (Fig. 2b and Supplementary Fig. 8). Specifically, compared to the corresponding subunit in PSI$_3$-IsiA$_{43}$, IsiA-i-1 and IsiA-o-1 in PSI$_1$-IsiA$_{13}$ shifted toward the lumen by 2-5 Å, whereas IsiA-i-6 and IsiA-o-7 moved toward the cytoplasm by 3-10 Å. These structural variations likely contribute to the differences in the binding modes and excitation-energy transfer (EET) pathways between PSI$_3$-IsiA$_{43}$ and PSI$_1$-IsiA$_{13}$ (see detailed analysis below).

## Interactions between the IsiA inner and outer rings
Our structural analysis showed that adjacent IsiA subunits within the same ring are tightly associated, similar to those in previously reported PSI$_3$-IsiA$_{18}$ structures[40–42], and are crucial for maintaining the double-ring architecture. In contrast, IsiA proteins belonging to the inner and outer rings predominantly form relatively weak, nonspecific hydrophobic interactions (Supplementary Fig. 9).

Upon analyzing the PSI$_3$-IsiA$_{43}$ complex, we observed that the 18 IsiA molecules in the inner ring displayed threefold symmetry, whereas the 25 IsiA subunits in the outer ring exhibited an asymmetrical arrangement, resulting in different interaction patterns between different inner-outer IsiA pairs (Supplementary Fig. 9c–t). Despite this heterogeneity, the interactions between the inner and outer IsiA of each pair predominantly occur at two similar patches located on the cytoplasmic and lumenal sides (Supplementary Fig. 9a, b). In most cases, lumenal interactions involve Phe200, Pro199, and Leu197 on the outer face of inner IsiAs, and Chl504, BCR521, and Ile297 on the inner side of the outer IsiAs. The cytoplasmic interactions are contributed by Chl516, Trp235, and Leu238 from the outer face of the inner IsiAs, and by Trp28 and Trp29 on the inner face of the outer IsiAs. Moreover,

Leu352 from the C-terminal helix of the adjacent IsiAs in the outer ring also contributes to cytoplasmic hydrophobic interactions. Furthermore, in four inner-outer IsiA pairs (i-2:o-1, i-4:o-4, i-7:o-8, i-15:o-19), a hydrogen bond is formed at the lumenal side between the sidechain of Gln194 of IsiA-i and the sidechain of Gln67 from IsiA-o (Supplementary Fig. 9d, f, i, q). In addition, a small number of specific hydrogen bond interactions were formed in other IsiA pairs, reinforcing the interactions between the inner and outer rings.

The core-IsiA and inner-outer IsiA interactions in the PSI$_1$-IsiA$_{13}$ complex differ slightly from those in PSI$_3$-IsiA$_{43}$, especially at the IsiA-i-6/IsiA-o-7 end, where the IsiA subunits are positioned closer to the core and exhibit vertical shifts (Supplementary Fig. 10). As a result, in the PSI$_1$-IsiA$_{13}$ structure, an extra hydrogen bond is formed between Glu349 from the C-terminal helix of IsiA-i-6 and Chl1218 of PsaB (Supplementary Fig. 10a), and additional hydrophobic interactions were observed between Trp28 of IsiA-i-5 and Phe310 and Phe311 of PsaB (Supplementary Fig. 10b). Overall, the IsiA-i-5 and IsiA-i-6 subunits of the inner ring bind more tightly to the PSI core than to those in the PSI$_3$-IsiA$_{43}$ supercomplex. Moreover, IsiA-o-7 also exhibits a tighter binding with IsiA-i-6, forming additional hydrophobic interactions and a hydrogen bond between Gln194 of IsiA-i-6 and Gln67 of IsiA-o-7 (Supplementary Fig. 10a). In addition to these horizontal movements within the membrane, IsiA-o-5, IsiA-o-6, and IsiA-o-7 in the PSI$_1$-IsiA$_{13}$ complex shift vertically, thereby altering their interaction surfaces with the inner layer. As a result, the associations within several inner IsiA-PsaB chlorophyll pairs differ from those observed in PSI$_3$-IsiA$_{43}$ (Supplementary Fig. 10c).

## Rotation and assembly of the outer ring
Previous studies have shown that the single IsiA ring exhibits organizational heterogeneity, characterized by positional rotations of the IsiA subunits relative to the trimeric PSI core in the PSI$_3$-IsiA$_{18}$ complex[45]. Here, we also investigated the heterogeneity of the IsiA layers in both the PSI$_3$-IsiA$_{43}$ and PSI$_1$-IsiA$_{13}$ complexes using multibody refinement in Relion[46] (Supplementary Fig. 11). We divided each of the two complexes into three bodies: the PSI core, inner IsiA layer, and outer IsiA layer (Supplementary Fig. 11a, c). When assessing PSI$_3$-IsiA$_{43}$, we found that the inner ring is stably associated with the PSI core, while the outer ring exhibits notable flexibility relative to both the core and the inner IsiA ring. Specifically, the outer ring shows a horizontal rotation of approximately 28.2 Å within the membrane plane but lacks vertical movement (Fig. 3a, b).

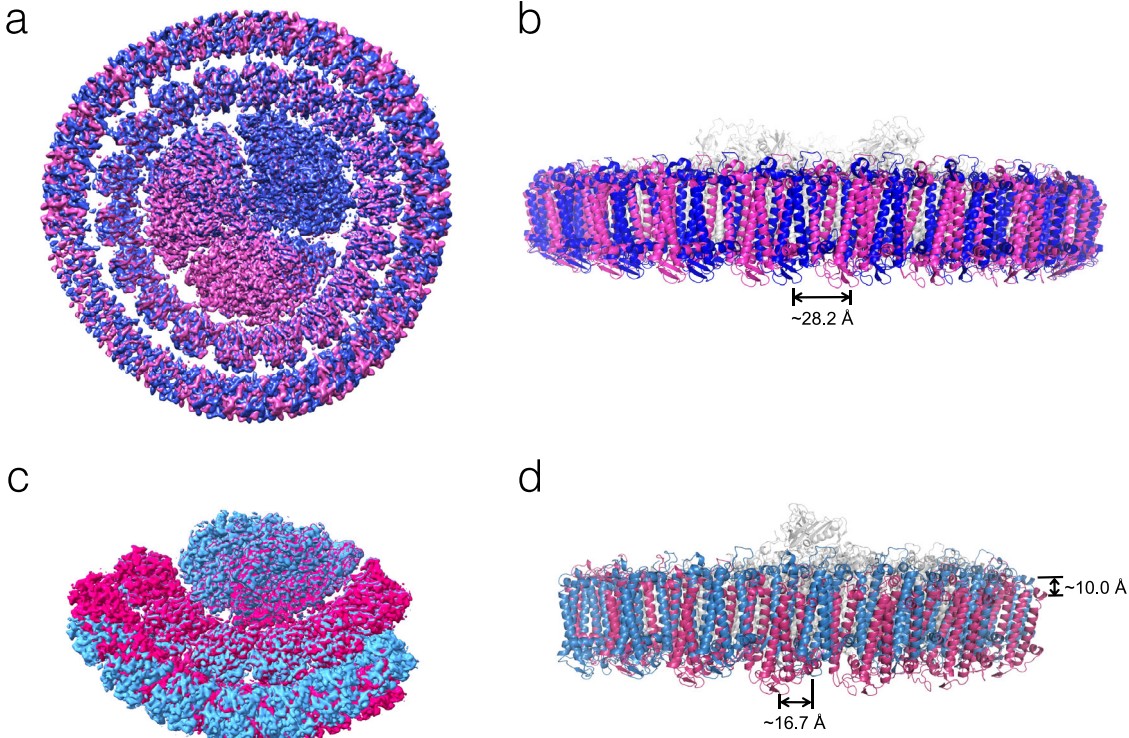

**Fig. 3 | Heterogeneity of PSI₃-IsiA₄₃ and PSI₁-IsiA₁₃ revealed by multibody refinement. a** Comparison of two density maps (magenta and blue) with the largest deviation in the top eigenvector from multibody refinement of PSI₃-IsiA₄₃, viewed from the cytoplasmic side. **b** Comparison of two structural models fitted into the two density maps of PSI₃-IsiA₄₃ in (**a**), viewed along the membrane plane. The PSI core and the inner ring are in gray, and the outer IsiAs are in magenta and blue. The shifting distance of the outer IsiAs between the two models is indicated at the bottom. **c** Comparison of two density maps (shown in red and sky blue) with the

largest deviation in the top eigenvector from the multibody refinement of PSI₁-IsiA₁₃, viewed from the cytoplasmic side. **d** Comparison of two structural models fitted into the two density maps of PSI₁-IsiA₁₃ in (**c**), viewed along the membrane plane. The PSI core and the inner ring are shown in gray, and the outer IsiAs are shown in red and sky blue. The maximal shifting distances of the outer IsiAs between the two models along both vertical and horizontal directions are indicated.

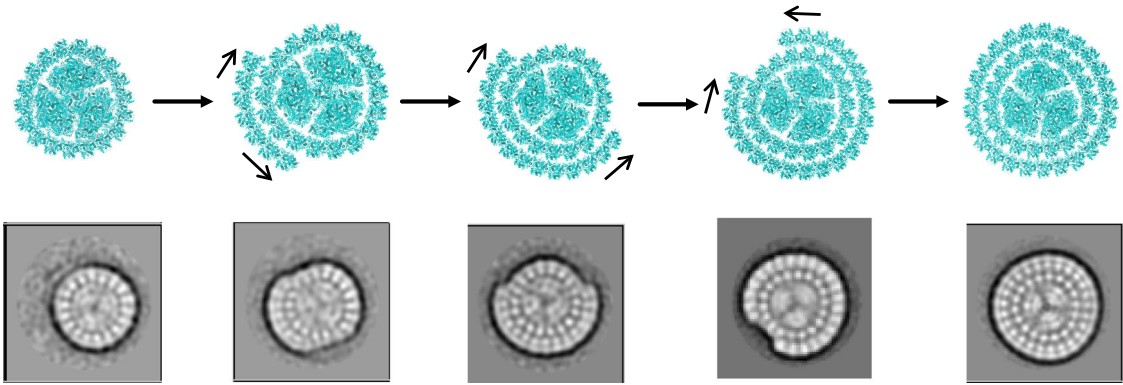

**Fig. 4 | Proposed assembly process of the PSI-IsiA double-ring supercomplex.** The 2D classification of negatively-stained PSI-IsiA particles (bottom) depicts the potential assembly process of the PSI₃-IsiA₄₃ complex (top). The assembly begins with the formation of the single-ring PSI₃-IsiA₁₈ complex. The outer IsiA subunits

then assemble around the PSI₃-IsiA₁₈ complex, either extending in one direction or simultaneously in both directions, as shown by black arrows. The size of the 2D classification box (bottom) is 500 × 500 Å.

Similar to PSI₃-IsiA₄₃, we found that PSI₁-IsiA₁₃ also has a relatively stable inner IsiA layer and a flexible outer layer relative to the PSI core but exhibits more complex motion patterns compared to PSI₃-IsiA₄₃ (Fig. 3c, d). The outer IsiA layer exhibited simultaneous fluctuations in both horizontal and vertical directions. In particular, the outer IsiA subunits of PSI₁-IsiA₁₃ show a horizontal shift of ~16.7 Å and a vertical shift of ~10.0 Å. Together, these results indicate that the outer IsiA layer in the monomeric PSI₁-IsiA₁₃ complex undergoes more pronounced fluctuations than the corresponding part in the trimeric PSI₃-IsiA₄₃ complex.

Intriguingly, our negative-staining electron microscopy results showed that several different classes of PSI-IsiA complexes are present in the purified PSI₃-IsiA₄₃ sample (Fig. 4). These complexes exhibit variations in the number of IsiA proteins incorporated into their outer ring, shedding light on the potential assembly mechanism of the double-ring PSI-IsiA complex. Our results suggest that the trimeric PSI core may initially assemble with a single IsiA ring containing 18 IsiA subunits. Subsequently, additional IsiA proteins might bind to the outside of the inner IsiAs in a continuous manner, either by adding in

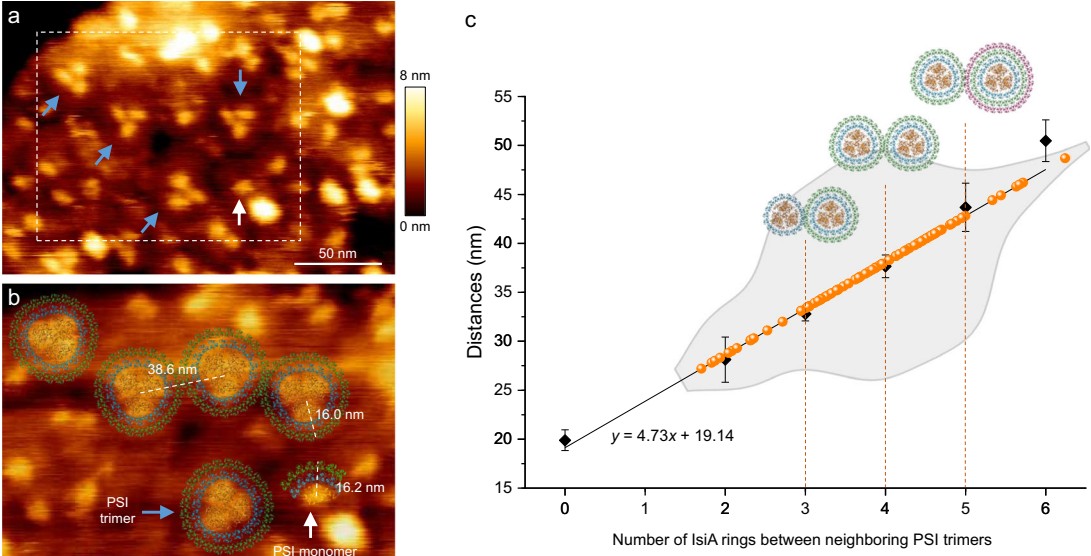

**Fig. 5 | AFM of native thylakoid membranes isolated from the iron-starved *T. elongatus* cells (Fe⁻ cell-2). a** AFM topographic overview of the thylakoid membrane fragment containing PSI-IsiA supercomplexes possessing the trimeric (blue arrows) and monomeric PSI core (white arrow). **b** Zoom-in view of the thylakoid membrane from the boxed area in (**a**). PSI-IsiA complexes with double IsiA rings and layers are annotated by superimposing the PSI$_3$-IsiA$_{43}$ (blue arrow) and PSI$_1$-IsiA$_{13}$ (white arrow) structures on the AFM image. The distances between the centers of two PSI trimers, between the center of a PSI trimer to its outer IsiA ring, and between one PSI monomeric protrusion to its outer IsiA layer are indicated. **c** Analysis of the distances between PSI trimers within the PSI-IsiA supercomplexes from different membrane patches (orange data points, *n* = 83), which have been

demonstrated to be closely correlated with the numbers of IsiA rings between the neighboring PSI trimers. Black data points and fitted curve were based on the previously established PSI-IsiA model[31]. Error bars on the data points represent SD = 1.06, 2.32, 0.72, 1.17, 2.47, 2.13 for the number of IsiA rings ranging from 0 to 6, respectively (*n* = 51). The gray-shaded area, represented as a violin plot derived from the distribution of the orange data points and superimposed on the fitted line, indicates the density distribution of orange data points. A large proportion of PSI-IsiA supercomplexes are shown to possess two IsiA rings, as indicated by three, four, or five IsiA surroundings between two neighboring PSI trimers (indicated by structural models).

## Membrane organization of PSI-IsiA complexes

To examine the PSI-IsiA assemblies in a context resembling their native state, we visualized their structures and lateral arrangement in thylakoid membranes isolated from iron-starved *T. elongatus* cells (Fe⁻ cell-2) using AFM (Fig. 5 and Supplementary Fig. 12). The trimeric PSI cores were clearly visible as high protrusions on the membrane surface (Fig. 5a, b). Moreover, we observed particles corresponding to the monomeric PSI core associated with double-layered IsiAs. The PSI$_3$-IsiA$_{43}$ and PSI$_1$-IsiA$_{13}$ structural models obtained from cryo-EM analysis were fitted nicely onto the AFM topograph (Fig. 5b and Supplementary Fig. 12), suggesting the cryo-EM structures likely reflect the natural physiological state of cyanobacterial PSI-IsiA under extreme iron deficiency.

Furthermore, we analyzed the distances between different PSI cores in adjacent PSI-IsiA supercomplexes, which have been demonstrated to be closely correlated with the number of IsiA rings in these supercomplexes[31]. Our analysis showed that, in most cases, the distances between neighboring PSI core trimers corresponded to three, four, or five IsiA layers, with four IsiA layers being the dominant form. This analysis suggests that a significant proportion of the PSI-IsiA supercomplexes possess two layers of IsiA rings in Fe⁻ cell-2 (Fig. 5c), and double-ring PSI-IsiA complexes appear to be prevalent in native thylakoid membranes from Fe⁻ cell-2 (Supplementary Fig. 12). Together, these findings highlight the fact that PSI-IsiA complexes with varying structures exist in native thylakoid membranes, consistent

with previous observations in *Synechococcus elongatus* PCC 7942 and *Synechococcus elongatus* UTEX 2973[31,44].

## Pigment arrangement and potential energy transfer pathways

Our structures revealed the detailed arrangement of chlorophyll molecules in the PSI$_3$-IsiA$_{43}$ and PSI$_1$-IsiA$_{13}$ complexes, illustrating the sophisticated pigment networks in PSI-IsiA assemblies, particularly at the interface between the inner and outer IsiA rings. Notably, 731 Chl *a* molecules are present in the double IsiA rings of the PSI$_3$-IsiA$_{43}$ structure, which is approximately three times the number of core pigments. Of the total Chl *a* molecules in IsiA subunits, 473 are located on the cytoplasmic side and 258 on the lumenal side of the thylakoid membrane. In addition, the Chl molecules exhibit a continuous and uniform distribution on the cytoplasmic side, whereas they form clusters on the lumenal side (Supplementary Fig. 13). The uneven distribution of pigments between the cytoplasmic and lumenal sides of thylakoid membranes was also observed in previously reported PSI$_3$-IsiA$_{18}$ structures[40–42].

The dense arrangement of Chl molecules results in the formation of multiple interfacial Chl pairs, in which the two Chl molecules belong to two adjacent protein subunits and are located in close proximity (Fig. 6a, b and Supplementary Fig. 14). These Chl pairs, particularly those at the interface between the inner and outer IsiA subunits, may facilitate rapid EET within the PSI-IsiA complexes. Although the interaction patterns of different inner-outer IsiA pairs vary, our structures identify common Chl molecules potentially involved in efficient EET from outer to inner IsiA subunits, including Chl 506, 516, and 517 on the outer face of inner-IsiA subunits, and Chl 504, 508, 510, and 519 on the inner face of outer-IsiA subunits. Moreover, Chl 519 of IsiA from the inner ring may also constitute the primary site for donating excitation energy to the PSI core, as elucidated in our previous report[41].

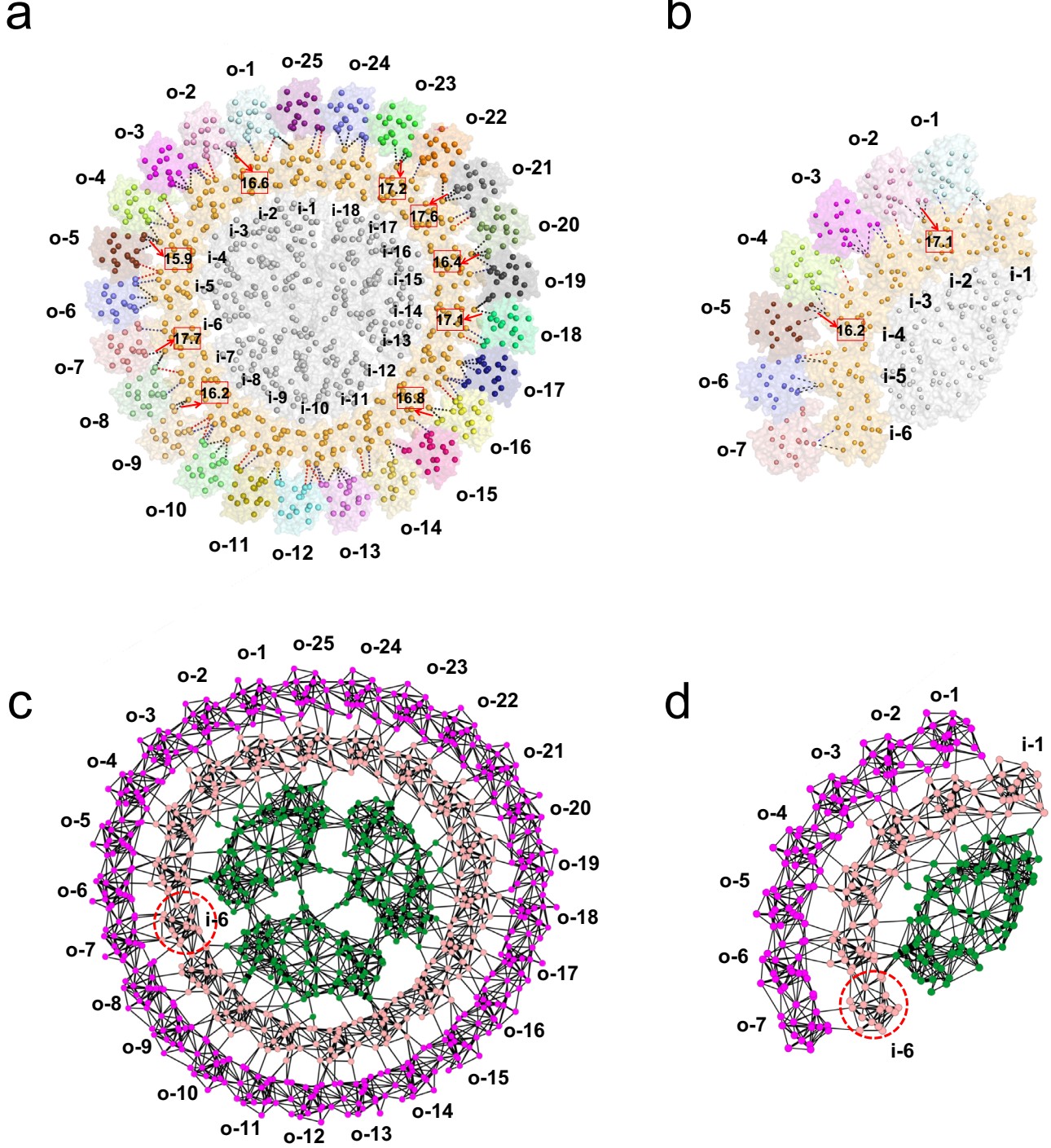

**Fig. 6 | Chlorophyll arrangement and proposed EET pathways within PSI$_3$-IsiA$_{43}$ and PSI$_1$-IsiA$_{13}$. a,b** The arrangement of Chl molecules in PSI$_3$-IsiA$_{43}$ (**a**) and PSI$_1$-IsiA$_{13}$ (**b**). Interfacial Chl molecules between inner and outer IsiA pairs are linked by dashed lines (≤23 Å Mg-Mg distance). Blue, black and red dashed lines link the Chl pairs containing Chl 506, 516 and 517 of the inner IsiAs, respectively. Distances under 18 Å are indicated with red arrows and highlighted by red boxes. The color codes of IsiA subunits are the same as in Fig. 1a. **c,d** Structure-based FRET calculations showing potential high-efficiency EET pathways in PSI$_3$-IsiA$_{43}$ (**c**) and PSI$_1$- IsiA$_{13}$ (**d**). Chl molecules are shown as spheres, and the FRET processes between neighboring chlorophylls with lifetime <40 ps are illustrated as black lines. The FRET processes with longer lifetime (≥ 40 ps) are omitted for clarity, and the data can be found in supplementary files. Line width indicates the relative FRET rate of each chlorophyll-chlorophyll pair (the wider the line, the faster the rate). The EET pathways involving IsiA-i-6 are highlighted with a red dashed circle. Source data for (**c**) and (**d**) are provided as a Source Data file.

To estimate the EET efficiency, we next calculated the Förster resonance energy transfer (FRET) rates of interfacial Chl pairs in the two PSI-IsiA structures (Fig. 6c, d and Supplementary Tables 2, 3). According to the Förster theory, FRET rates strongly depend on the Mg-Mg distances and mutual dipole moment orientations between two adjacent chlorophylls. Although the limited resolutions (3.4–3.5 Å) of our structures do not allow a precise determination of these structural parameters, the position and orientation of each chlorophyll molecule in IsiAs are similar to those in previously reported PSI-IsiA structures[40–43] (Supplementary Fig. 15). Therefore, we suggest that

while the calculated results may be biased regarding several individual EET pathways, the overall EET networks calculated based on our structures are largely reliable. Our analysis shows that the EET within the same IsiA ring/layer is highly efficient because of the presence of multiple rapid transfer pathways between two adjacent IsiA subunits. Moreover, in the $PSI_3$-$IsiA_{43}$ complex, the major EET pathways from outer to inner IsiAs are primarily achieved through the Chl pairs outer508-inner506, outer508-inner516, outer504-inner517, outer519-inner517, and outer510-inner516 (Supplementary Table 3), consistent with our structural observation. In addition, our data indicate that EET is highly effective on the luminal side, even with a smaller number of Chl molecules, in agreement with a previous report[40]. Interestingly, we found that while efficient EET occurs in each inner-outer IsiA pair, the IsiA-i-6 in the trimeric complex may exhibit reduced efficiency in the transfer of excitation energy to the core (Fig. 6c and Supplementary Table 3). As a result, energy absorbed by IsiA-i-6 may first be transferred to neighboring IsiAs within the same ring, and then further transferred to the core. Furthermore, we calculated the FRET rates of interfacial Chl pairs using the $PSI_3$-$IsiA_{43}$ structure in which the outer IsiA ring undergoes a large-scale rotation in our multibody analysis (Fig. 3a, b and Supplementary Table 4). The results showed that the primary interfacial Chl pairs involved in the rapid outer-to-inner IsiA EET are similar to those in the original $PSI_3$-$IsiA_{43}$ complex, albeit belonging to distinct IsiA pairs (Supplementary Tables 3, 4). These analyses suggested that the rotational shift of the outer IsiA ring does not result in a significant change in the efficiency of outer-to-inner IsiA EET. This finding is consistent with previous results obtained for the $PSI_3$-$IsiA_{18}$ complex[45], showing that a robust EET from the IsiA antennae to the core occurs despite the large-scale rotation of the IsiA ring.

The inward shift toward the PSI core and vertical movement of the terminal IsiA subunits in the $PSI_1$-$IsiA_{13}$ supercomplex lead to slight differences in the potential energy transfer pathways compared to $PSI_3$-$IsiA_{43}$, particularly at the IsiA-i-6/IsiA-o-7 end (Fig. 6d). The closer association between IsiA-i-6 and the core enables Chl 519 of IsiA-i-6 to transfer excitation energy to PsaB more effectively than the corresponding pathways in $PSI_3$-$IsiA_{43}$ (Fig. 6c, d and Supplementary Tables 2, 3). In contrast, EET between IsiA-o-7 and IsiA-i-6 in $PSI_3$-$IsiA_{43}$ is highly efficient, whereas the $PSI_1$-$IsiA_{13}$ complex shows fewer effective EET pathways (shorter than 40 ps) in the corresponding region (Fig. 6c, d), presumably due to the vertical shift of the outer IsiAs. Despite these differences, our structural and FRET analyses demonstrate that in various PSI-IsiA complexes, the excitation energy absorbed by IsiA proteins is rapidly equilibrated within the IsiA ring/layer and further transferred from the outer IsiAs to the inner IsiAs and the cores through multiple pathways. This is consistent with a recent study showing that multiple chlorophyll molecules in multiple IsiAs contribute together to the rapid energy transfer within PSI-IsiA complexes[36]. The intricate energy transfer network observed in our structures enables effective and robust EET in both PSI-IsiA complexes, even under changing iron-limited conditions.

## Discussion

In this study, we determined the cryo-EM structures of two types of PSI-IsiA complexes, $PSI_3$-$IsiA_{43}$ and $PSI_1$-$IsiA_{13}$, purified from iron-starved *T. elongatus* BP-1 cells. While previous structural studies reported various complexes of PSI core associated with a single IsiA ring/layer[23,33,40–43], our $PSI_3$-$IsiA_{43}$ structure reveals a large cyanobacterial photosynthetic apparatus of over 3 MDa, comprising a trimeric PSI core encircled by a closed double-layered IsiA assembly (Fig. 1). In comparison, $PSI_1$-$IsiA_{13}$ contains approximately one-third of the $PSI_3$-$IsiA_{43}$ complex, exhibiting an arrangement similar to the PSI-moiety-1 of $PSI_3$-$IsiA_{43}$. However, the terminal IsiA subunits (IsiA-i-5 ~ 6 and IsiA-o-6 ~ 7) in $PSI_1$-$IsiA_{13}$ exhibit both horizontal and vertical shifts compared to the corresponding subunits in $PSI_3$-$IsiA_{43}$ (Fig. 2).

A major distinction between the two complexes lies in the curvature of their membrane-spanning regions (Supplementary Fig. 8). In $PSI_3$-$IsiA_{43}$, the curvature is nearly flat, while in $PSI_1$-$IsiA_{13}$, the IsiA region tilts relative to the core. This curvature might be related to the native architecture of cyanobacterial thylakoid membranes, which are relatively flat with locally curved regions[45,47]. This suggestion is consistent with earlier reports showing that several PSI complexes exhibit curved architecture and have been suggested to influence the morphology and topological organization of thylakoid membranes[8,45,48,49]. We propose that the double-ring architecture may provide enhanced stabilization of mesoscale membrane curvature, while the monomeric complex state may reflect a more flexible arrangement. In our AFM imaging, we observed the co-existence of both monomeric and trimeric PSI-IsiA complexes within the same region (Fig. 5 and Supplementary Fig. 12), consistent with previous studies showing the presence of multiple types of PSI-IsiA complexes in cyanobacterial thylakoid membranes[31,33,38]. The heterogeneous organization of PSI-IsiA complexes observed in previously reported studies, as well as our AFM analyses, may contribute to the shaping and stabilization of the membrane architecture while maintaining a certain degree of flexibility.

In addition, the curved architecture of $PSI_1$-$IsiA_{13}$ may be related to the positional shift of the terminal IsiA subunits (Fig. 2). The stronger interactions of these terminal IsiA subunits with the core in $PSI_1$-$IsiA_{13}$ may affect the binding of other IsiA proteins. Additionally, the $PSI_1$-$IsiA_{13}$ complex lacks the PsaL subunit, which is crucial for PSI trimer and tetramer formation[41,43]. Thus, the loss of PsaL and the positional shift of IsiA subunits may prevent the trimeric PSI-IsiA assembly, resulting in the formation of the monomeric $PSI_1$-$IsiA_{13}$ complex. Consistently, our negative staining electron microscopy results suggest that the $PSI_3$-$IsiA_{43}$ complex may not be formed through assembly of three individual PSI-IsiA monomers. Instead, its assembly process might involve the initial formation of a trimeric $PSI_3$-$IsiA_{18}$ complex as a crucial intermediate step (Fig. 4). The ring-shaped IsiA architecture prevents the inward and vertical shift of inner IsiAs relative to the core, thereby providing a docking platform for the sequential addition of outer IsiA proteins and formation of a closed outer ring. This stepwise model differs from the simple monomer-to-trimer assembly model proposed for $PSI_3$-$IsiA_{18}$[42], suggesting highly flexible and dynamic assembly modes of different PSI-IsiA complexes[32,38].

Our structural analysis further indicates that the consistent orientation of IsiA subunits enables tight interactions between adjacent IsiA subunits within the same ring, potentially resulting in a strong tendency of IsiA to form a ring-shaped supercomplex. This suggestion is consistent with an earlier report showing that in the cyanobacterium lacking PsaF and PsaJ subunits, the trimeric PSI core is still associated with an IsiA ring composed of 17 IsiA proteins[50]. However, the IsiA-IsiA interactions also show some extent of flexibility, allowing horizontal and vertical shifts and hence the formation of different PSI-IsiA assemblies[33,38,39] (Fig. 2). Furthermore, we show that the interactions between the inner ring and the PSI core, as well as between the two rings, are relatively weak (Supplementary Fig. 9). This organization likely enables the dynamic regulation of IsiA ring binding within PSI-IsiA complexes, consistent with our multibody calculations demonstrating that the outer IsiA layer in both PSI-IsiA complexes exhibits considerable heterogeneity in spatial positioning and rotational flexibility (Fig. 3). The flexible PSI-IsiA interactions were also reported in an earlier study[45], and is likely the key structural basis for the formation of various PSI-IsiA complexes and the self-assembled IsiA supercomplexes without PSI core[33,38].

In addition, the specific orientation of IsiA subunits also assists in optimizing the arrangement of pigment molecules in PSI-IsiA complexes, particularly the chlorophylls (Chls 516-519) absent in CP43. These Chls are positioned on the inner and outer surfaces of the IsiA rings, and are involved in forming the primary EET pathways within the

complexes, as shown by our FRET rate calculations (Supplementary Tables 2-4). The previous report[32] as well as our absorption spectra and fluorescence emission kinetics experiments (Supplementary Figs. 3b, c) together indicate that the double-ring PSI-IsiA complex performs highly efficient excitation-energy capture and transfer processes, which largely depend on the chlorophyll arrangement determined by the specific orientation of IsiA subunits. In addition, while we found that the outer IsiA ring of $PSI_3$-$IsiA_{43}$ complex exhibits a large-scale rotation, resulting in changes of relative positions of each outer-inner IsiA pair in $PSI_3$-$IsiA_{43}$ complex, our FRET analysis (Supplementary Tables 3, 4) revealed that the Chl pairs involved in the outer-to-inner IsiA EETs are conserved, albeit belonging to different IsiA pairs. Therefore, the outer ring rotation may fine-tune but not significantly affect the outer-to-inner IsiA EET efficiency. This suggestion is consistent with recent structural and ultrafast spectroscopic analyses of PSI-IsiA complexes, showing that although heterogeneous energy transfer was observed, the whole IsiA-PSI supercomplexes exhibit very rapid EET[36]. These features exemplify a design principle of photosynthetic antenna systems, in which the EET efficiency is dynamically regulated while maintaining a high robustness in response to environmental fluctuations, thereby enhancing the adaptability of the system.

Previous AFM analysis of cyanobacterial thylakoid membranes revealed that $PSI_3$-$IsiA_{18}$ complexes accumulate under iron-starved conditions[31,44]. While our AFM imaging across multiple fields of view and biological replicates (Fig. 5 and Supplementary Fig. 12) showed the enrichment of $PSI_3$-$IsiA_{43}$ as well as the presence of other types of PSI-IsiA complexes within the membranes, suggesting that cyanobacterial PSI prefers to bind to a double-layered IsiA under extreme iron deficiency. Consistent with this, our biochemical and structural results demonstrated that more IsiA proteins are induced in cells under more severe iron starvation (Supplementary Fig. 1), and the PSI complexes with single- and double-layered IsiA subunits are present primarily in moderately and severely iron-deficient cells, respectively (Supplementary Fig. 2). Under iron-replete conditions, IsiA protein levels decrease, and the PSI trimers without IsiA become the major population (Supplementary Figs. 1, 2). On the basis of these findings, we hypothesize that the formation of the double IsiA ring constitutes a mechanism employed by cyanobacteria to successfully adapt to extremely low-iron environments. In comparison, the single IsiA ring is most likely the result of adaptation to moderately low-iron conditions. The structural reversibility of PSI and different types of PSI-IsiA complexes revealed in the present study demonstrates that the assembly of PSI-IsiA complexes is regulated in a dynamic and adaptable manner rather than a rigid process in response to different iron-starvation conditions.

In summary, by integrating biochemistry, cryo-EM, AFM, and FRET analyses, we provided extensive data highlighting the heterogeneity and dynamics of the PSI-IsiA supercomplexes, underscoring the adaptive strategies utilized by cyanobacteria to optimize photosynthesis under environmental stresses.

## Methods

### Induction, purification and characterization of $PSI_3$-$IsiA_{43}$ and $PSI_1$-$IsiA_{13}$ complexes

*T. elongatus* was cultured at 50 °C in BG-11 medium with continuous bubbling under white light at ~ 50 μmol photons m$^{-2}$ s$^{-1}$. The Fe$^+$ cells were cultured in the BG-11 medium containing a standard concentration of Fe$^{2+}$ (0.021 mM). To induce the accumulation of IsiA proteins, the Fe$^+$ cells were transferred to the iron-free BG-11 medium at two different dilution ratios (1/25 and 1/60, v/v), resulting in Fe$^-$ cell-1 (moderate iron limitation) and Fe$^-$ cell-2 (severe limitation). The Fe$^-$ cell-2 was further cultured in iron-supplemented medium for five days to obtain the Fe$^{-/+}$ cells (iron restored after severe limitation). The absorption spectra of cells grown under iron-deficient conditions were

measured to monitor IsiA protein accumulation. Cells were harvested by centrifugation at 5000 × g for 5 min, when the $Q_y$ absorption peak shifted from 683 to 676 nm. To obtain thylakoid membranes, the cell pellet was resuspended in a buffer containing 20 mM HEPES, pH 7.5, 20 mM NaCl, 10 mM CaCl$_2$, 10 mM MgCl$_2$ and disrupted using a high-pressure homogenizer at 750 bar. Unbroken cells were pelleted by centrifugation at 2500 × g for 5 min, and the resulting supernatant was subjected to ultracentrifugation at 70,000 × g for 40 min to collect the thylakoid membranes[41].

To purify the $PSI_3$-$IsiA_{43}$ and $PSI_1$-$IsiA_{13}$ complexes, thylakoid membranes were resuspended in 20 mM HEPES (pH 7.5), 10 mM CaCl$_2$, and 10 mM MgCl$_2$ at 0.7 mg Chl per ml. The membranes were then solubilized with 1% (w/v) dodecyl-$\beta$-d-maltoside ($\beta$-DDM, Anatrace) for 20 min on ice. The solubilized membranes were then loaded onto the 0.1–1.5 M sucrose density gradient in a buffer containing 20 mM HEPES pH7.5, 10 mM CaCl$_2$ and 10 mM MgCl$_2$ and 0.03% (w/v) dodecyl-$\alpha$-d-maltoside ($\alpha$-DDM, Anatrace), following centrifugation at 154,390 × g for 19 h using a SW41 rotor (Beckman) at 4 °C. The $PSI_3$-$IsiA_{43}$ and $PSI_1$-$IsiA_{13}$ complexes were concentrated using Amicon Ultra Centrifugal Filters (Millipore) with 100 kDa cut-off, to final concentrations of 3 and 2 mg ml$^{-1}$ (in Chl), respectively, for cryo-EM specimen preparation.

The room-temperature absorption spectra of *T. elongatus* cells, PSI and PSI-IsiA complexes were measured using a spectrophotometer (Hitachi, Japan). Pigment composition was analyzed using an HPLC system (Shimadzu, Japan). Samples were mixed with cold acetone (final concentration of 80% (v/v)). The mixture was vortexed and then centrifuged at 13,000 g for 10 min to extract pigments. The resulting supernatant was injected to a C-18 reversed-phase column (Shim-pack GIST C18, Shimadzu), eluted at a flow rate of 1 mL/min at room temperature, and monitored at 440 nm. The elution profile was as follows: From 0 to 1 min, 100% buffer A containing 87% (v/v) acetonitrile and 10% (v/v) methanol was used. From 1 to 20 min, a linear gradient of buffer A decreasing from 100 to 0% and buffer B (Methanol: Hexane = 8:2) increasing from 0% to 100% was applied. From 20 to 23 min, 100% buffer B was used. From 23 to 24 min, a linear gradient of buffer B was applied, decreasing from 100% to 0%. Finally, from 24 to 28 min, 100% buffer A was restored. Individual pigments were identified by the absorption spectrum of each elution peak.

P700 oxidation was measured as a transient decrease in absorption at 705 nm using a Joliot-type spectrophotometer (Bio-Logic SAS JTS-10). All samples were diluted into the same chlorophyll concentration, and 3 mM sodium ascorbate (electron donor) and 1 mM methyl-viologen (electron acceptor) were added to each sample. Two actinic lights of 630 nm (10 nm FWHM, 20 uE) and 720 nm (10 nm FWHM, 400 uE) were used for the measurement. The PSI oxidation kinetics were fitted with a mono-exponential model.

The protein composition of *T. elongatus* cells, the purified $PSI_3$-$IsiA_{43}$ and $PSI_1$-$IsiA_{13}$ complexes were analyzed using 4–20% (w/v) gradient sodium dodecyl sulfate polyacrylamide gel electrophoresis (SDS-PAGE). The gel was stained with Coomassie Brilliant Blue R250. The band potentially corresponding to the IsiA protein was excised and digested with trypsin overnight. The digested sample was analyzed on a nanoLC Q-Exactive mass spectrometer (Thermo Fisher Scientific) in line with an Easy-nLC 1000 HPLC system (Thermo Fisher). MS/MS spectra from each LC-MS/MS run were searched against the *Thermosynechococcus elongatus* BP-1 sequence using the Proteome Discoverer (Thermo Fischer Scientific version 1.4.0.288) searching algorithm with the following criteria: Full tryptic specificity was required, two missed cleavages were allowed, precursor ion mass tolerance was set to 10 ppm, fragment ion mass tolerance was 0.02 Da, and peptide FDR (false discovery rate) was set to 1%.

### Grid preparation and data acquisition

Aliquots (3 μL) of the sample ($PSI_3$-$IsiA_{43}$ or $PSI_1$-$IsiA_{13}$) were placed onto a freshly glow-discharged holey carbon grid (GIG-C311) and

blotted for 4 s at a blotting force of level 3 using Vitrobot Mark IV (Thermo Fisher). The grids were then vitrified by plunging into liquid ethane and stored in liquid nitrogen.

PSI$_3$-IsiA$_{43}$ grids were imaged using a 300 kV Titan Krios microscope (Thermo Fisher) equipped with a K2 direct electron detector (Gatan). Movies (32 frames per movie file) were captured at a defocus range of −1.5 to −2.5 μm at 130,000× magnification in super-resolution mode. The pixel size and total dose were 0.5 Å and 60 e$^-$ Å$^{-2}$. Data for the PSI$_1$-IsiA$_{13}$ samples were collected using a 200 kV Talos Arctica microscope (Thermo Fisher) equipped with a K2 direct detector (Gatan). Data were obtained at a defocus range of −1.2 to −2.5 μm, with 130,000× magnification and a pixel size of 0.52 Å.

### Image processing

Beam-induced motion in the raw movies was corrected using MotionCor 2.1[51]. During PSI$_3$-IsiA$_{43}$ image processing, the contrast transfer function (CTF) parameters of summed images were estimated using Gctf[52]. Image processing was further performed using Relion 3.1[46]. Approximately 2000 particles were manually picked from 1800 images and subjected to 2D classification. The best classes were selected and used as references for the reference-based autopicking. After particle picking, 2D classification was performed, and high-quality classes were selected. A total of 310,779 selected particles were used for 3D classification. After 3D classification, 95,950 particles from the class exhibiting a complete IsiA double ring were selected for refinement. The PSI$_3$-IsiA$_{43}$ complex was finally constructed at a resolution of 3.4 Å.

Image processing of PSI$_1$-IsiA$_{13}$ followed the same steps as that of PSI$_3$-IsiA$_{43}$. After excluding the false-positive and false-negative particles through 3D classification, the selected class containing 55,670 particles was constructed at 4.0 Å resolution. The map was subsequently optimized by CTF refinement and Bayesian polishing, resulting in a final resolution of 3.5 Å. The local resolutions of the final maps were calculated using ResMap. The workflows of the cryo-EM analysis of PSI$_3$-IsiA$_{43}$ and PSI$_1$-IsiA$_{13}$ are shown in Supplementary Figs. 4, 5, and the data collection and processing statistics are summarized in Supplementary Table 1.

### Model building and refinement

The cryo-EM structure of PSI$_3$-IsiA$_{18}$ from *Synechococcus elongatus* PCC 7942 (PDB code 6KIG)[41] and its single IsiA structure were initially docked onto the PSI$_3$-IsiA$_{43}$ map using UCSF Chimera for model building. Subsequently, the amino acid sequences were mutated to their counterparts in *T. elongatus*, and the model was manually adjusted using COOT[53] and refined iteratively using Phenix[54]. The same method was employed for model building of the PSI$_1$-IsiA$_{13}$ structure. The structural geometries were evaluated using MolProbity[55], with statistical details summarized in Supplementary Table 1. Figures were prepared using Chimera[56] and PyMOL (Molecular Graphics Systems, LLC).

### Multibody refinement and analysis

To explore the flexibility of IsiA subunits in the PSI$_3$-IsiA$_{43}$ and PSI$_1$-IsiA$_{13}$ complexes, refined particles from individual datasets were used for 3D multibody refinement in Relion[46]. These datasets were generated after CTF refinement and Bayesian polishing procedures. For PSI$_3$-IsiA$_{43}$ multibody refinement, three rigid bodies corresponding to the PSI core moiety, inner IsiA ring, and outer IsiA ring were assigned by individual soft masks. For PSI$_1$-IsiA$_{13}$, three rigid bodies were defined as the monomeric PSI core moiety, the inner six IsiAs, and the outer seven IsiAs, using individual soft masks. The initial angular sampling rate, offset range, and offset step were set to 1.5°, six pixels, and 1 pixel, respectively. In the subsequent flexibility analysis using the Relion _flex_ analyze program, the motion of the three rigid bodies was decomposed along 18 eigenvectors. The component along the first top eigenvector, comprising ten maps (state001-state010), was output. The translation extents were visualized by superposing the corresponding states.

### AFM measurement of thylakoid membranes

Thylakoid membranes were isolated from *T. elongatus* cells as previously reported[31,57–59] and described below. *T. elongatus* cells (OD$_{750}$ ~ 0.8) were harvested by centrifugation at 5000 g for 10 min at 4 °C. Cell pellets were washed with a buffer containing 50 mM HEPES-NaOH, pH 7.5, 30 mM CaCl$_2$, and 1% Protease Inhibitor Cocktail, and resuspended in a buffer containing 50 mM HEPES-NaOH, pH 7.5, 30 mM CaCl$_2$, and 800 mM sorbitol. The cells were then broken down using glass beads (150-212 μm) at 4 °C. To obtain pure thylakoid membranes, we separated the membrane fractions in a stepwise sucrose gradient (2.0, 1.5, 1.3, 1.0, 0.5 M) and centrifugation at 40,000 rpm in a SW41 Ti Swinging-Bucket Rotor for 1 h at 4 °C. The Chl-enriched band at the 1.3–1.5 M sucrose interface was collected for AFM imaging.

Thylakoid membranes (2 μL) were adsorbed onto freshly cleaved mica surfaces using 38 μL adsorption buffer (10 mM Tris-HCl, pH 7.5, 150 mM KCl, and 25 mM MgCl$_2$) and were incubated for 1 h at room temperature. After adsorption, the sample was gently rinsed with an imaging buffer (10 mM Tris-HCl, pH 7.5, 150 mM KCl) to remove non-adsorbed thylakoid membranes. High-resolution imaging was performed in liquid at room temperature using dynamic AFM tapping mode (AC mode) on a JPK BioAFM (Bruker, USA) equipped with an ULTRA S scanner and Ultra-Short Cantilever probe (k = 0.3 N/m, NanoWorld). The cantilevers were driven at a resonance frequency of 130–140 kHz, and AFM imaging was conducted with a target (setpoint) amplitude of 1.2 V. Scanning was performed at 3 Hz scanning frequency and 512 × 512 pixel resolution. The tip spring constant was routinely calibrated.

AFM images were processed and analyzed using JPK SPM Data Processing software (v 6.1.198). The center-to-center distances between adjacent PSI trimeric protrusions were measured from multiple membrane patches obtained from different samples. These measurements are shown as orange data points in Fig. 5c. For comparison, black data points and a fitted curve were generated based on the previously established model that correlates the center-to-center distances between PSI trimers with the number of surrounding IsiA rings. The distribution of the measured distances is illustrated by a violin plot (gray area shown in Fig. 5c), indicating the frequency of different values[31,44].

### FRET analysis

FRET analysis was performed based on the standard FRET theory[7]. The rate constant ($k_{FRET}$) was calculated using the equation $k_{FRET} = (CK^2) / (n^4 R^6)$. Here, $C$ is derived from the spectral overlap integral between the donor and acceptor Chls, $K$ is the dipole orientation factor, $n$ represents the refractive index, and $R$ is the distance between the central magnesium atoms of the two Chls. For energy transfer between Chl $a$ molecules, $C$ was set to 32.26, and $n$ was taken as 1.55, as estimated in ref. 60. The dipole orientation factor ($K^2$) is calculated as $K^2 = [\hat{u}_D \cdot \hat{u}_A - 3(\hat{u}_D \cdot R_{DA})(\hat{u}_A \cdot R_{DA})]^2$. Here, $\hat{u}_D$ and $\hat{u}_A$ are unit vectors representing the transition dipoles of the donor and acceptor Chls, derived from the coordinates of NB and ND atoms. $R_{DA}$ is the unit vector pointing from the donor to the acceptor Chl's central magnesium atoms. FRET rates were estimated computationally using Kim's algorithm implemented in Python (v3.8) and visualized with the interactive platform Gephi[61]. All calculated $K^2$ and $k_{FRET}$ values for individual chlorophyll pairs are provided in Supplementary files. Fast EET pathways (<20 ps) calculated using different PSI-IsiA structures are listed in Supplementary Tables 2-4.

**Reporting summary**

Further information on research design is available in the Nature Portfolio Reporting Summary linked to this article.

## Data availability

The atomic coordinates of the PSI$_3$-IsiA$_{43}$ and PSI$_1$-IsiA$_{13}$ complexes have been deposited in the Protein Data Bank (PDB) under accession codes 9LZJ (for PSI$_3$-IsiA$_{43}$) and 9LZK (for PSI$_1$-IsiA$_{13}$), respectively. The cryo-EM maps have been deposited in the Electron Microscopy Data Bank (EMDB) under accession codes EMD-63527 (for PSI$_3$-IsiA$_{43}$) and EMD-63528 (for PSI$_1$-IsiA$_{13}$). All other relevant data generated in this study are provided in the Supplementary Information and source data files. The source data underlying Fig. 6c, d, and Supplementary Figs. 1a and 3a are provided as a Source Data file. Source data are provided with this paper.

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

## Acknowledgements

Cryo-EM datasets were collected at the Center for Biological Imaging (CBI), Core Facilities for Protein Science at the Institute of Biophysics (IBP), Chinese Academy of Sciences (CAS). We thank B. Zhu, L. Chen, X. Huang, X. Li, G. Ji, D. Fan, T. Niu, F. Sun, and other staff members at the CBI (IBP, CAS) for technical support in EM data collection; and L. Niu, M. Zhang, and F. Yang for mass spectrometry. We thank Y. Zhu and X. Song from the Large-scale Instruments and Equipments Sharing Platform of Beijing University of Technology, and W. Liu from College of Chemistry and Life Science for technical assistance. We thank C. Zhang from the Plant Science Facility of the Institute of Botany, CAS, for support with the P700 assay. We thank Dr. Torsten Juelich from University of Chinese Academy of Sciences for the linguistic assistance during the preparation of this manuscript. The project was funded by the National Natural Science Foundation of China (31930064 to M.L., 32070259 to P.C., 32070109 to L.-N.L., and 32011530168 to M.L.), Royal Society (IEC\NSFC\191600 and URF\R\180030 to L.-N.L.), Biotechnology and Biological Sciences Research Council (BB/W001012/1, BB/V009729/1, BB/Y01135X/1, and BB/Y008308/1 to L.-N.L.), and Liverpool-Chinese Scholarship Council PhD studentship (Y.Z.).

## Author contributions

M.L., P.C., and L.-N.L. conceived and supervised the project; L.S. performed the sample preparation and characterization, cryo-EM data collection and processing; X.S., X.Z., and X.A. assisted in sample preparation and characterization; P.C. and L.S. performed model building, refinement, and structural analysis; Y.Z. and L.-N.L. performed AFM measurements and analyzed the data; P.C., L.S., L.-N.L., and M.L. wrote the manuscript; all authors discussed and commented on the results and the manuscript.

## Competing interests

Authors declare no competing interests.
