## [Transparent Peer Review file · Nature Communications]

Structural basis for the assembly and energy transfer between the cyanobacterial PSI core and the double-layered IsiA proteins

Corresponding Author: Professor Mei Li

Version 0:

Reviewer comments:

Reviewer #1

(Remarks to the Author)

Si et al. provide valuable insights into the mechanisms by which cyanobacteria adapt to iron-limited conditions through dynamic restructuring of the PSI-IsiA supercomplex. The high-resolution cryo-electron-microscopy structures of PSI₃-IsiA₄₃ and PSI₁-IsiA₁₃, along with the confirmation of their presence in native membranes via atomic force microscopy and the assessment of energy transfer efficiency through Förster resonance energy transfer (FRET) analysis, collectively demonstrate the critical role of these supercomplexes in optimizing light harvesting and energy transfer. In particular, the formation of a double-layered IsiA ring appears to represent an adaptive strategy for coping with extreme iron deficiency, while the structural flexibility of the outer IsiA layer emerges as a key feature facilitating efficient energy transfer under varying environmental conditions, as highlighted by the authors.

The findings presented in this manuscript are compelling. However, several ambiguities remain in the description of the mechanism and the results of the analysis that should be addressed before the manuscript can be considered for publication.

1. The observed blue shift of the Q_y absorption peak is used as an indirect indicator of increased IsiA expression; however, the evidence presented is insufficient to conclusively support this assumption. To more reliably assess IsiA expression levels, quantitative analyses using mass spectrometry or other suitable biochemical methods are necessary.
2. In the Discussion, the authors hypothesize that the formation of the double IsiA ring serves as a mechanism by which cyanobacteria adapt to severely iron-deficient environments. To reinforce this hypothesis, it would be valuable to include experimental data collected under conditions where iron levels are restored following deficiency. Demonstrating structural reversibility upon iron repletion would allow for a more definitive correlation between iron availability and supercomplex architecture.
3. Given the 3.4–3.5 Å resolution, it is difficult to determine inter-chlorophyll distances (e.g., Mg–Mg) and the orientation of diphenyl groups with Å-level precision. Nevertheless, the FRET analysis and the interpretation presented in Figure 6 appear to rely on structural parameters beyond the resolution limit. It would be appropriate for the authors to acknowledge this limitation in the text and clarify the extent to which their conclusions are sensitive to such fine structural details.
4. In FRET analysis, the dipole orientation factor (K^2) is calculated based on structural information. The calculated K^2 and rate constant (k_{FRET}) values for all chlorophyll pairs should be provided in the Supplementary Information for transparency and reproducibility.
5. On page 14, the authors state that “the rotational shift of the outer IsiA ring has little effect on the efficiency of outer-to-inner IsiA EET.” However, according to Förster theory, EET efficiency is highly sensitive to the relative orientation of transition dipole moments, which raises doubts about this conclusion. The authors should provide quantitative data or simulations demonstrating that the reported rotational shift truly has minimal impact on EET efficiency.

(Remarks to the Author)

This study provides a large amount of novel data about the interesting topic of the membrane complexes produced by cyanobacteria under iron starvation or high intensity light exposure. This is a topic of great interest to the photosynthesis community. This seems to be the first high-resolution structure of the double-ring structures that IsiA can form around PSI - which is a novel finding. There is a clear comparison of the newly obtained structures of PSI3-IsiA43 and PSI1-IsiA13, with a good level of detailed descriptions in the text with carefully constructed main-text figures, supported with appropriate supplementary figures. The AFM data and FRET analysis is crucial for extending the scope of the work and provides valuable new insights into the isiA-PSI system, however, these two aspects lack depth and could be improved by simply by providing further detail, as noted below. I think the work is of high quality and wide interest and I would be supportive of publication after the issues listed below have been addressed.

There are some significant issues that should be addressed -

- The analysis of AFM images including the measurements of centre-to-centre distances between PSI complexes, in Figure 5, is well done. The method is consistent with high-quality measurements from other AFM studies, including the authors' previous work. This is a valuable analysis and gives real insight into the relationship between the cryoEM structures of IsiA-PSI (extracted complexes) and the in-membrane structures of IsiA-PSI. However, the AFM analysis lacks depth and certainty and it could be improved upon.
- It is disappointing that only one field-of-view is shown (one AFM image, Fig 5A and a zoomed in version in Fig 5B). This is a weakness of the paper. Ideally, a gallery of several AFM images from different membrane patches from one biological sample and/or from different samples should be shown in the supplementary information and referred to in the main text. This is important to show that the data presented in the main text is representative of a typical membrane and not just one example. Any supplementary AFM data provided would not need to be of the high resolution and it would be acceptable if the quality was lower so long as the location of the protrusions of PSI trimers can be defined.
- There should be more clarity over how the graph in Figure 5C was generated. It is unclear what each orange datapoint represents (one single distance measurement from AFM data?), or the black datapoints (measurements from the models?) or the error bars of the black datapoints (provide n=? And SD=?), or the grey shaded area of the graph. It is also whether this graph represents measurements from only one AFM image (Figure 5A) or from several AFM images. Currently there is no information about how these measurements were made in either the figure caption or the Methods; there is only a citation of the authors' previous AFM studies, ref 31 and 44. This omission is an oversight and the authors should be given the opportunity to rectify this. I suggest that this information should be briefly stated in the figure caption with any further detail provided in the Methods.
- The FRET analysis of Figure 6C/D is interesting and valuable because it puts the structural data into the energetic context. The analysis may be of high quality, but unfortunately some information is missing and it lacks depth. It is not clear what each grey line between pigment sites represents. The figure caption states "showing potential high-efficiency EET pathways (lifetime < 40 ps)" which could mean that ALL pigment couples with a grey line drawn between them have calculated FRET with a time constant (lifetime) calculated as below 40 ps or it could mean that only some of these connected pigments have this lifetime. I suggest that the author explicitly state in the figure caption which FRET couples are displayed in Figure 6C/D.
- Another weakness of the FRET analysis is that this type of display of the FRET pairs is very binary - either some connection or zero connection. It would be better if the variability of the FRET rates between different pigments could be visualized. For example, the lines between pigments could be colour coded based on how high or low their lifetime is. Or the thickness of the bond between pigments could be varied to represent the FRET rate. This would provide more information to readers about how the EET connections within this complex. I assume the authors have this data because the calculations have been performed already so this should not take much more effort to provide further visualizations, and it would benefit readers. This could be another panel in Figure 6 and/or an additional figure in the supplementary information.
- The Discussion section provides an inadequate level of referencing. There should be clear comparison against the state of the art by citation of appropriate literature - currently there only two references in the entire discussion. There should be several in every paragraph - to provide evidence or context to each claim that is made. Further sentences could be added to the discussion of the structural findings to compare to published works on (i) membrane curvature, (ii) protein subunit assembly, (iii) isiA subunit orientation, (iv) iron starvation.
- It is also disappointing that there is no mention in the discussion section of the significance and wider implications of the findings related to the AFM data or the findings related to the FRET analysis. Of course the focus of this study was the cryoEM structures but it would benefit readers if there was a significant discussion about the how these findings provide new insight about the organization of these complexes within native thylakoid membranes and energy transfer processes within. This is a significant oversight and the authors should be able to do this quite easily.

Minor comments -

- The claim about how IsiA assembles around PSI, related to figure 4, is stated as too much of a certainty because EM data shows a static snapshot of the protein complexes and provides no information about the kinetic progression (assembly over time). It is possible that figure 4 shows fragmentation of a larger complex into smaller ones due to the purification process. Or it is possible that the intermediate size complexes are a common features of the membranes as shown by the AFM data. The two sentences below should be reworded to with some qualifies like "may/ might" and the other possibilities should be noted/ discussed. "Our results indicate that the trimeric PSI core initially assembles with a single IsiA ring containing 18 IsiA subunits. Subsequently, additional IsiA proteins bind to the outside of the inner IsiAs in a continuous manner, either by adding in one direction or extending in both directions, until a closed outer ring is formed (Fig. 4)."
- minor formatting errors in reference list
- "the Qy absorption band" - should clarify to "the Qy absorption band of chlorophyll a" (line 113 and 128)

- should label the other bands in the figure of the sucrose gradients (Fig S1c) to give context with the other thylakoid proteins.
- should show a magnified version of the Qy peak of purified isiA-PSI complexes (Fig S2b) to show the peak shift because it is not currently clear. For example, a graph of only 650-700 nm over the same width as the current x-axis.
- In the Methods - Should specify how the purified isiA-PSI was concentrated (which brand of spin concentrators? or pelleting + resuspension?) because this can change protein quality.
- The Discussion should cite figures, as appropriate, to direct readers to the associated evidence.

Reviewer #3

(Remarks to the Author)

Iron limitation is a common stress factor for cyanobacteria. To cope with this, they produce a protein called IsiA, which helps them harvest light and protect the photosystems. IsiA forms rings around PSI cores. In this study, PSI-IsiA complexes from *Thermosynechococcus elongatus* were analysed using cryo-EM. The PSI3-IsiA43 shows a trimeric PSI core surrounded by two IsiA rings. The other (PSI1-IsiA13) has a monomeric core with fewer IsiAs. Atomic force microscopy also showed how these complexes are arranged in membranes. The authors claim this results help to understand how cyanobacteria adjust light harvesting and energy transfer when iron is missing.

The work is technically sound and the resolution is okay to make the claims the authors do. I however completely lack an interpretation of the physiological relevance or any further characterisation that indicates there might be a benefit or function that goes beyond "it's an curiosity of nature". As it stands, the study remains a descriptive structural analysis, and apart from the double-ring architecture, the complex does not differ substantially from previously published PSI-IsiA structures. Without further data addressing physiological function or photophysical properties, I believe the manuscript may be more appropriate for a specialized structural biology journal in its current form.

Version 1:

Reviewer comments:

Reviewer #1

(Remarks to the Author)

The authors' transparency is commendable and substantially strengthens the manuscript. However, in reviewing the supplementary files, I noted that the units of the calculated quantities, particularly the rate constant, are not explicitly specified. For clarity and reproducibility, please add the appropriate units to the column headers (e.g., ps⁻¹). Once this minor revision is addressed, I will be pleased to recommend the manuscript for acceptance.

Reviewer #2

(Remarks to the Author)

The authors have comprehensively addressed all of my comments. I recommend publication in the current form.

Reviewer #3

(Remarks to the Author)

The authors improved the interpretation of their results and highlighted more clearly that what they observe is of physiological relevance. I have no more objections to publish the improved version of this article.

Response to Reviewer #1:

Si et al. provide valuable insights into the mechanisms by which cyanobacteria adapt to iron-limited conditions through dynamic restructuring of the PSI-IsiA supercomplex. The high-resolution cryo-electron-microscopy structures of PSI₃-IsiA₄₃ and PSI₁-IsiA₁₃, along with the confirmation of their presence in native membranes via atomic force microscopy and the assessment of energy transfer efficiency through Förster resonance energy transfer (FRET) analysis, collectively demonstrate the critical role of these supercomplexes in optimizing light harvesting and energy transfer. In particular, the formation of a double-layered IsiA ring appears to represent an adaptive strategy for coping with extreme iron deficiency, while the structural flexibility of the outer IsiA layer emerges as a key feature facilitating efficient energy transfer under varying environmental conditions, as highlighted by the authors.

The findings presented in this manuscript are compelling. However, several ambiguities remain in the description of the mechanism and the results of the analysis that should be addressed before the manuscript can be considered for publication.

Responses: We thank the reviewer for the positive comments and valuable suggestions.

1. The observed blue shift of the Q_y absorption peak is used as an indirect indicator of increased IsiA expression; however, the evidence presented is insufficient to conclusively support this assumption. To more reliably assess IsiA expression levels, quantitative analyses using mass spectrometry or other suitable biochemical methods are necessary.

2. In the Discussion, the authors hypothesize that the formation of the double IsiA ring serves as a mechanism by which cyanobacteria adapt to severely iron-deficient environments. To reinforce this hypothesis, it would be valuable to include experimental data collected under conditions where iron levels are restored following deficiency. Demonstrating structural reversibility upon iron repletion would allow for a more definitive correlation between iron availability and supercomplex architecture.

Responses: Thank you very much for these excellent suggestions. We agree with the reviewer

that the evidence for using Q_y absorption peak as an indicator of increased IsiA expression is insufficient. Therefore, to roughly quantify IsiA expression levels, we cultured *T. elongatus* cells under different conditions (iron concentrations) and obtained several cell types, namely Fe^+ cell (cultured under iron-sufficient conditions), Fe^- cell-1 (cultured under moderate iron-limiting conditions), Fe^- cell-2 (cultured under severe iron-limiting conditions) and $Fe^{-/+}$ cell (Fe^- cell-2 further cultured under iron-supplemented conditions). We measured the absorption spectra of these cells and performed electrophoresis analysis (SDS-PAGE) of the cell extracts. The results are shown in Fig. S1 in the revised manuscript (also see below). We found that the IsiA expression levels increased as the iron concentration in the culture medium decreased (Fig. S1a), and the Q_y absorption peak of cells gradually shifted from 683 nm to 678.5 nm and finally to 676 nm (Fig. S1b). Moreover, when the iron concentrations were restored after iron deficiency, the Q_y absorption peak of $Fe^{-/+}$ cells shifted back to 681.5 nm, and the IsiA protein levels decreased significantly, similar to those of the Fe^+ cells. These data collectively support our suggestion that the Q_y absorption peak could be used as an indicator of increased IsiA expression, in good agreement with previous studies (refs 31, 32, 44).

Supplementary Figure 1. Iron-dependent IsiA accumulation and reversibility of *T. elongatus* cells. (a) SDS-PAGE analysis of the cell extracts of different types of *T. elongatus* cells treated with various iron conditions. The Coomassie-stained band corresponding to IsiA indicated by an arrow was confirmed through mass spectrometry. (b) Room-temperature absorption spectra of cells grown under different iron conditions:

Fe⁻ cell-1 (moderate limitation), Fe⁻ cell-2 (severe limitation), Fe^{-/+} (iron restored after severe limitation), and Fe⁺ (sufficiency). The Q_y region is shown enlarged.

Furthermore, we solubilized and further fractionated the thylakoid membranes purified from Fe⁻ cell-1, Fe⁻ cell-2 and Fe^{-/+} cells by sucrose density gradient ultracentrifugation, and then characterized the major PSI bands using negative-stain electron microscopy. The results are shown in Fig. S2 in the revised manuscript (see below). We found that Fe⁻ cell-1 primarily formed the PSI₃-IsiA₁₈ complex (Fig. S2b), whereas Fe⁻ cell-2 mainly accumulated PSI₁-IsiA₁₃ and PSI₃-IsiA₄₃ complexes (Fig. S2c, d). In contrast, Fe^{-/+} cells only formed PSI₃ complexes (Fig. S2e), consistent with our electrophoresis analysis results which indicated limited IsiA levels in Fe^{-/+} cells. These data demonstrate the structural reversibility of PSI and PSI-IsiA complexes under varying iron concentrations, supporting our proposed correlation between iron availability and supercomplex architecture. We have included these newly obtained results and modified the text in the revised manuscript.

Supplementary Figure 2. Characterization of the isolated supercomplexes cultured under different iron concentrations. (a) Sucrose density gradient ultracentrifugation of thylakoid membranes isolated from

T. elongatus cells grown under different iron conditions: Fe⁻ cell-1 (moderate limitation), Fe⁻ cell-2 (severe limitation), and Fe^{-/+} (iron restored after severe limitation). Major bands are labeled and the PSI bands (bold labels) were collected for negative staining analysis. (b-e) Negative staining images and 2D classification results of PSI complexes obtained from Fe⁻ cell-1 (b), Fe⁻ cell-2 (c, d), and Fe^{-/+} cell (e).

3. Given the 3.4–3.5 Å resolution, it is difficult to determine inter-chlorophyll distances (e.g., Mg–Mg) and the orientation of diphenyl groups with Å-level precision. Nevertheless, the FRET analysis and the interpretation presented in Figure 6 appear to rely on structural parameters beyond the resolution limit. It would be appropriate for the authors to acknowledge this limitation in the text and clarify the extent to which their conclusions are sensitive to such fine structural details.

Responses: Thank you for your suggestion. We have modified the text to acknowledge the limitation and clarify that the FRET rates calculated based on our structures may be biased for certain individual EET pathways. Despite that, we believe that the overall EET networks calculated based on our structures are largely reliable. We compared the chlorophyll arrangement of IsiA proteins from different PSI-IsiA structures, as shown in Fig. S14 in the revised manuscript (see below). Structural superimposition showed that the position and orientation of all chlorophyll molecules in IsiAs from different structures are similar. We have modified the text as following: *According to the Förster theory, FRET rates strongly depend on the Mg-Mg distances and mutual dipole moment orientations between two adjacent chlorophylls. Although the limited resolutions (3.4-3.5 Å) of our structures do not allow a precise determination of these structural parameters, the position and orientation of each chlorophyll molecule in IsiAs are similar to those in previously reported PSI-IsiA structures⁴⁰⁻⁴³ (Fig. S14). Therefore, we suggest that while the calculated results may be biased regarding several individual EET pathways, the overall EET networks calculated based on our structures are largely reliable.*

Supplementary Figure 14. Structural comparison of IsiA protein among different PSI-IsiA complexes.

(a-b) Side view (a) and Top view (b) of IsiA subunits from PSI-IsiA complexes of different cyanobacterial species. IsiA subunits in the PSI₃-IsiA₄₃ complex from *T. elongatus* (this study, pink) and the PSI₃-IsiA₁₈ complexes from *T. vulcanus* (PDB code: 6K33, orange), *Synechococcus* sp. PCC 7942 (PDB code 6KIG, green), and *Synechocystis* sp. PCC 6803 (PDB code 7UMH, blue) were aligned.

4. In FRET analysis, the dipole orientation factor (K^2) is calculated based on structural information. The calculated K^2 and rate constant (k_{FRET}) values for all chlorophyll pairs should be provided in the Supplementary Information for transparency and reproducibility.

Responses: Thanks for this suggestion. We have provided the complete output files of the FRET calculation for both PSI₁-IsiA₁₃ and PSI₃-IsiA₄₃, containing K^2 and rate constant (k_{FRET}) values, as additional supplementary files (named FRET-rate-monomer-output.csv and FRET-rate-trimer-output.csv). Moreover, we have summarized the FRET pairs with rapid excitation transfer (lifetime less than 20 ps) from chlorophylls in outer IsiAs to those in inner IsiAs and from chlorophylls in inner IsiAs to those in the PSI core in both complexes, as shown in Tables S2-4 in our revised manuscript.

5. On page 14, the authors state that “the rotational shift of the outer IsiA ring has little effect on the efficiency of outer-to-inner IsiA EET.” However, according to Förster theory, EET efficiency is highly sensitive to the relative orientation of transition dipole moments, which

raises doubts about this conclusion. The authors should provide quantitative data or simulations demonstrating that the reported rotational shift truly has minimal impact on EET efficiency.

Responses: We thank the reviewer for raising this important issue. Following the reviewer's suggestion, we have calculated the FRET rates of interfacial Chl pairs using the PSI₃-IsiA₄₃ structure, in which the outer IsiA ring undergoes a large-scale rotation in our multibody analysis, and summarized the FRET pairs with rapid excitation transfer (lifetime less than 20 ps) from chlorophylls in outer IsiAs to those in inner IsiAs and from chlorophylls in inner IsiAs to those in the PSI core (Table S4 in the revised manuscript). The results showed that the primary interfacial Chl pairs involved in the rapid outer-to-inner IsiA EET are similar to those in the original PSI₃-IsiA₄₃ complex, albeit belonging to different IsiA pairs. For example, rapid FRET rates (lifetime less than 20 ps) from Chl 508 (outer IsiA) to Chl 516 (inner IsiA) calculated based on one PSI₃-IsiA₄₃ structure are present in o-3/i-3, o-6/i-5, o-10/i-8, o-13/i-10, o-17/i-13, o-21/i-16, o-24/i-18 IsiA pairs (Table S3 in the revised manuscript), whereas rapid FRET rates for the same Chl pair calculated based on another PSI₃-IsiA₄₃ structure occur in o-1/i-1, o-5/i-4, o-8/i-6, o-12/i-9, o-15/i-11, o-19/i-14, o-22/i-16, o-23/i-17 IsiA pairs (Table S4 in the revised manuscript). These analyses suggest that the rotational shift of the outer IsiA ring did not lead to significant changes in the outer-to-inner IsiA EET efficiency, although the underlying mechanism remains to be further explored. We have included these data in our revised manuscript. We recognize that the calculations are not based on accurate structures, and therefore, we have toned down the relevant statements.

Response to Reviewer #2:

This study provides a large amount of novel data about the interesting topic of the membrane complexes produced by cyanobacteria under iron starvation or high intensity light exposure. This is a topic of great interest to the photosynthesis community. This seems to be the first high-resolution structure of the double-ring structures that IsiA can form around PSI - which is a novel finding. There is a clear comparison of the newly obtained structures of PSI3-IsiA43 and PSI1-IsiA13, with a good level of detailed descriptions in the text with carefully constructed main-text figures, supported with appropriate supplementary figures. The AFM data and FRET analysis is crucial for extending the scope of the work and provides valuable new insights into the isiA-PSI system, however, these two aspects lack depth and could be improved by simply by providing further detail, as noted below. I think the work is of high quality and wide interest and I would be supportive of publication after the issues listed below have been addressed.

Responses: We appreciate the extremely positive feedback and thoughtful suggestions to our manuscript.

There are some significant issues that should be addressed -

- The analysis of AFM images including the measurements of centre-to-centre distances between PSI complexes, in Figure 5, is well done. The method is consistent with high-quality measurements from other AFM studies, including the authors' previous work. This is a valuable analysis and gives real insight into the relationship between the cryoEM structures of IsiA-PSI (extracted complexes) and the in-membrane structures of IsiA-PSI. However, the AFM analysis lacks depth and certainty and it could be improved upon.
- It is disappointing that only one field-of-view is shown (one AFM image, Fig 5A and a zoomed in version in Fig 5B). This is a weakness of the paper. Ideally, a gallery of several AFM images from different membrane patches from one biological sample and/or from different samples should be shown in the supplementary information and referred to in the

main text. This is important to show that the data presented in the main text is representative of a typical membrane and not just one example. Any supplementary AFM data provided would not need to be of the high resolution and it would be acceptable if the quality was lower so long as the location of the protrusions of PSI trimers can be defined.

Response: This is an excellent suggestion. In the revised manuscript, we have included two additional AFM topographs, which were obtained from different membrane patches and biological replicates, as shown in Fig. S11 in our revised manuscript (see below). These AFM images show individual PSI monomeric and trimeric protrusions in thylakoid membranes and the potential arrangements of double-ring and single-ring PSI-IsiA assemblies based on their lateral distances, in agreement with the observations depicted in Fig. 5A.

Supplementary Figure 11. AFM topographs of native thylakoid membranes isolated from the iron-

starved *T. elongatus* cells (Fe⁻ cell-2) revealing the arrangement of PSI-IsiA supercomplexes. (a, b) Two different membrane patches. Right panels: PSI-IsiA complexes with double IsiA rings/layers and single IsiA rings are annotated by superimposing the PSI₃-IsiA₄₃, PSI₃-IsiA₁₈ (PSI₃-IsiA₄₃ with the outer IsiA ring removed), and PSI₁-IsiA₁₃ structures on the AFM image.

- There should be more clarity over how the graph in Figure 5C was generated. It is unclear what each orange datapoint represents (one single distance measurement from AFM data?), or the black datapoints (measurements from the models?) or the error bars of the black datapoints (provide n=? And SD=?), or the grey shaded area of the graph. It is also whether this graph represents measurements from only one AFM image (Figure 5A) or from several AFM images. Currently there is no information about how these measurements were made in either the figure caption or the Methods; there is only a citation of the authors' previous AFM studies, ref 31 and 44. This omission is an oversight and the authors should be given the opportunity to rectify this. I suggest that this information should be briefly stated in the figure caption with any further detail provided in the Methods.

Response: We agree with the reviewer's comments. In the revised manuscript, we have updated the Methods section and figure legends to clarify the figure and our approach of data analysis. Specifically, the orange data points represent the center-to-center distances between adjacent PSI trimeric protrusions measured from multiple AFM topographs (including Fig. 5 and new Fig. S11) obtained from different membrane patches in three independent samples. The black points and the fitted black line correspond to the predicted distances based on the previously published correlation between IsiA ring numbers and PSI spacing (ref. 31); error bars indicate the standard deviation. The grey shaded area represents a violin plot illustrating the overall distribution and density of the measured distances, related to the orange data points. We have also added detailed descriptions of these measurements in the revised Fig. 5 legend (see below, the changes are highlighted in yellow) and Methods section (Line 576-584 in the revised manuscript).

Fig. 5 legend

(c) Analysis of the distances between PSI trimers within the PSI-IsiA supercomplexes from different membrane patches (orange data points, n=83), which have been demonstrated to be closely correlated with the numbers of IsiA rings between the neighbouring PSI trimers. Black data points (n=51; SD=1.06, 2.32, 0.72, 1.17, 2.47, 2.13 for the number of IsiA rings ranging from 0 to 6, respectively) and fitted curve based on the previously established PSI-IsiA model³¹. The grey shaded area reflects the distribution density of orange data points, highlighting that most complexes contain two IsiA rings. A large proportion of PSI-IsiA supercomplexes are shown to possess two IsiA rings, as indicated by three, four, or five IsiA surroundings between two neighboring PSI trimers (indicated by structural models).

- The FRET analysis of Figure 6C/D is interesting and valuable because it puts the structural data into the energetic context. The analysis may be of high quality, but unfortunately some information is missing and it lacks depth. It is not clear what each grey line between pigment sites represents. The figure caption states "showing potential high-efficiency EET pathways (lifetime < 40 ps)" which could mean that ALL pigment couples with a grey line drawn between them have calculated FRET with a time constant (lifetime) calculated as below 40 ps or it could mean that only some of these connected pigments have this lifetime. I suggest that the author explicitly state in the figure caption which FRET couples are displayed in Figure 6C/D.

- Another weakness of the FRET analysis is that this type of display of the FRET pairs is very binary - either some connection or zero connection. It would be better if the variability of the FRET rates between different pigments could be visualized. For example, the lines between pigments could be colour coded based on how high or low their lifetime is. Or the thickness of the bond between pigments could be varied to represent the FRET rate. This would provide more information to readers about how the EET connections within this complex. I assume the authors have this data because the calculations have been performed already so this should not take much more effort to provide further visualizations, and it would benefit readers. This could be another panel in Figure 6 and/or an additional figure in the supplementary information.

Responses: Thank you very much for your valuable suggestions. We have re-produced Figs.

6c, 6d by changing the thickness of lines between pigments to represent the FRET rates (thicker lines indicate faster FRET rates). In addition, we modified the Fig. 6 legend to clarify that only pigment pairs with calculated lifetimes less than 40 ps are connected by black lines (see below, the changes are highlighted in yellow). In addition, we calculated FRET rates for all pigment pairs, and the data are provided in additional supplementary files.

Fig. 6 legend

(c, d) Structure-based FRET calculations showing potential high-efficiency EET pathways in PSI₃-IsiA₄₃ (c) and PSI₁-IsiA₁₃ (d). Chl molecules are shown as spheres, and the FRET processes between neighboring chlorophylls with lifetime < 40 ps are illustrated as black lines. The FRET processes with longer lifetime (≥ 40 ps) are omitted for clarity, and the data can be found in supplementary files. Line width indicates the relative FRET rate of each chlorophyll-chlorophyll pair (the wider the line, the faster the rate). The EET pathways involving IsiA-i-6 are highlighted with a red dashed circle.

- The Discussion section provides an inadequate level of referencing. There should be clear comparison against the state of the art by citation of appropriate literature - currently there only two references in the entire discussion. There should be several in every paragraph - to provide evidence or context to each claim that is made. Further sentences could be added to the discussion of the structural findings to compare to published works on (i) membrane curvature, (ii) protein subunit assembly, (iii) isiA subunit orientation, (iv) iron starvation.

- It is also disappointing that there is no mention in the discussion section of the significance and wider implications of the findings related to the AFM data or the findings related to the FRET analysis. Of course the focus of this study was the cryoEM structures but it would benefit readers if there was a significant discussion about the how these findings provide new insight about the organization of these complexes within native thylakoid membranes and energy transfer processes within. This is a significant oversight and the authors should be able to do this quite easily.

Responses: We thank the reviewer for the constructive comments. Based on these suggestions,

we have expanded the Discussion section and cited additional references in the revised manuscript. Specifically, we added further sentences and citations regarding (i) thylakoid membrane curvature and the photosynthetic supercomplex architecture (line 381-394, refs 8, 45, 47, 48, 49), (ii) the dynamic assembly and structural variability of PSI-IsiA complexes (line 407-410, refs 32, 38, 41, 42, 43), (iii) the role of IsiA subunit orientation in complex assembly and energy transfer (line 413-418, 424-426, 430-447, refs 32, 33, 36, 38, 39, 50), and (iv) cyanobacterial adaptation to iron starvation (line 448-458, refs 31, 44). In the revised Discussion section, we compare our findings with recent studies, and discuss the significance and wider implications of our AFM and FRET results. In particular, our AFM results link the structural data to the native state of the PSI-IsiA complexes in thylakoid membranes, collectively suggesting the correlation between photosynthetic complex formation and the membrane curvature. Our multibody refinement and FRET rate calculations collectively demonstrate that PSI-IsiA complexes are dynamically assembled while exhibiting robust energy transfer networks, reflecting an adaptive design principle of cyanobacterial light-harvesting systems.

Minor comments -

- The claim about how IsiA assembles around PSI, related to figure 4, is stated as too much of a certainty because EM data shows a static snapshot of the protein complexes and provides no information about the kinetic progression (assembly over time). It is possible that figure 4 shows fragmentation of a larger complex into smaller ones due to the purification process. Or it is possible that the intermediate size complexes are common features of the membranes as shown by the AFM data. The two sentences below should be reworded to with some qualifies like "may/ might" and the other possibilities should be noted/ discussed. "Our results indicate that the trimeric PSI core initially assembles with a single IsiA ring containing 18 IsiA subunits. Subsequently, additional IsiA proteins bind to the outside of the inner IsiAs in a continuous manner, either by adding in one direction or extending in both directions, until a closed outer ring is formed (Fig. 4)."

Response: We thank the reviewer for pointing this out. We have modified the text in the revised manuscript according to the reviewer's suggestion as following: *Our results suggest that the trimeric PSI core may initially assemble with a single IsiA ring containing 18 IsiA subunits. Subsequently, additional IsiA proteins might bind to the outside of the inner IsiAs in a continuous manner, either by adding in one direction or extending in both directions, until a closed outer ring is formed (Fig. 4). Alternatively, the observed intermediates may reflect a native feature of incomplete PSI-IsiA complexes within the membranes, or could be a result of disassembly of larger complexes into smaller ones during purification.*

- minor formatting errors in reference list

Response: Corrected. Thank you.

- "the Q_y absorption band" - should clarify to "the Q_y absorption band of chlorophyll a" (line 113 and 128)

Response: Corrected. Thank you.

- should label the other bands in the figure of the sucrose gradients (Fig S1c) to give context with the other thylakoid proteins.

Responses: Following the reviewer's suggestion, we have labeled all bands in Fig. S1c (now Fig. S2a in the revised version).

- should show a magnified version of the Q_y peak of purified isiA-PSI complexes (Fig S2b) to show the peak shift because it is not currently clear. For example, a graph of only 650-700 nm over the same width as the current x-axis.

Responses: Thanks for your suggestion. We have provided a magnified version of the Q_y peak shown in Fig. S2b (now Fig. S3b in the revised manuscript), according to the reviewer's suggestion.

- In the Methods - Should specify how the purified isiA-PSI was concentrated (which brand of spin concentrators? or pelleting + resuspension?) because this can change protein quality.

Responses: Thanks for your suggestion. The purified PSI-IsiA complexes were concentrated using Amicon Ultra Centrifugal Filters (Millipore) with 100 kDa cut-off, to final concentrations of 2-3 mg ml⁻¹ (in Chl). We have provided this information in the Methods section.

- The Discussion should cite figures, as appropriate, to direct readers to the associated evidence.

Responses: Thanks for your suggestion. We have cited appropriate figures in the Discussion section.

Response to Reviewer #3:

Iron limitation is a common stress factor for cyanobacteria. To cope with this, they produce a protein called IsiA, which helps them harvest light and protect the photosystems. IsiA forms rings around PSI cores. In this study, PSI-IsiA complexes from *Thermosynechococcus elongatus* were analysed using cryo-EM. The PSI3-IsiA43 shows a trimeric PSI core surrounded by two IsiA rings. The other (PSI1-IsiA13) has a monomeric core with fewer Isias. Atomic force microscopy also showed how these complexes are arranged in membranes. The authors claim this results help to understand how cyanobacteria adjust light harvesting and energy transfer when iron is missing.

The work is technically sound and the resolution is okay to make the claims the authors do. I however completely lack an interpretation of the physiological relevance or any further characterisation that indicates there might be a benefit or function that goes beyond "it's an curiosity of nature". As it stands, the study remains a descriptive structural analysis, and apart from the double-ring architecture, the complex does not differ substantially from previously published PSI-IsiA structures. Without further data addressing physiological function or photophysical properties, I believe the manuscript may be more appropriate for a specialized structural biology journal in its current form.

Responses: We thank the reviewer for the highly positive comments on the quality of our structures and the endorsement of our statements. We agree that the PSI core and the inner IsiA moieties of the complex do not differ substantially from previously published PSI-IsiA structures, which we have mentioned in our manuscript (Fig. S5a/S6a in the original/revised version). However, we would like to point out that this is the first high-resolution structures reporting the PSI core bound with double-layered IsiA subunits and, to the best of our knowledge, the first study that integrates the structural information, AFM data, and FRET rate calculations for PSI-IsiA complexes.

Our AFM results link the structural data to the native state of the PSI-IsiA complexes in

thylakoid membranes, collectively suggesting the correlation between PSI-IsiA complex formation and the membrane curvature. Our multibody refinement and FRET rate calculations collectively show that PSI-IsiA complexes are dynamically assembled while exhibiting robust EET, which might be a key mechanism for cyanobacteria to maintain efficient photosynthesis in changing environments. We believe that our findings provide an important basis for understanding the physiological functions of PSI-IsiA complexes.

Furthermore, following the reviewers' suggestions, we characterized *T. elongatus* cells and PSI-IsiA complexes obtained under different iron concentrations using biochemical and structural methods. Our results indicate the correlation between the iron concentration, absorption spectra of cells, and IsiA protein levels. Moreover, our negative staining results strongly suggest that the PSI core and IsiA proteins reversibly assemble and disassemble in response to changes in iron accessibility, which might be an important adaptive mechanism for cyanobacteria to cope with the iron-limited stress.

Integrating these newly obtained results, we have expanded the discussion in the revised manuscript, including the assembly and organization of complexes within native thylakoid membranes, the correlation between iron accessibility and structural reversibility of various PSI(-IsiA) complexes, the relationship between complex formation and membrane curvature, and the structural dynamics of PSI-IsiA complexes and robust energy transfer processes.

We believe that our findings provide mechanistic insights into the assembly and structural variations of PSI core-antenna complexes, which will not only advance current understanding of photosynthetic complexes but also their reprogramming to design of photosynthetic modules for biofuel/bioenergy applications, which may attract a broad attention from researchers in photosynthetic, synthetic biology, applied biotechnology, protein design coupled with AI technologies.

RESPONSE TO REVIEWERS' COMMENTS

Reviewer #1 (Remarks to the Author):

The authors' transparency is commendable and substantially strengthens the manuscript. However, in reviewing the supplementary files, I noted that the units of the calculated quantities, particularly the rate constant, are not explicitly specified. For clarity and reproducibility, please add the appropriate units to the column headers (e.g., ps⁻¹). Once this minor revision is addressed, I will be pleased to recommend the manuscript for acceptance.

Response: Thank you for your positive comments and suggestion. The units of lifetime and FRET rate are picosecond (ps) and ps⁻¹, respectively, as previously reported (Jan P. Götze & Heiko Lokstein, "Excitation Energy Transfer between Higher Excited States of Photosynthetic Pigments: 1. Carotenoids Intercept and Remove B Band Excitations". *ACS Omega* 2023. DOI: 10.1021/acsomega.3c05895).

We have checked Supplementary Tables 2-4 in our revised manuscript to ensure that the appropriate units have been added to the column headers, and that these units are consistent with those reported in recently published studies involving the FRET rate calculations, which are listed below.

- 1) Sheng, X., *et al.*, *Nature Plants*, 2019. DOI: 10.1038/s41477-019-0543-4. Extended Data Figure 9.
- 2) Ishii, A., *et al.*, *eLife*, 2023. DOI: 10.7554/eLife.84488. Table 5.
- 3) Shen, L., *et al.*, *Science Advance*, 2024. DOI: 10.1126/sciadv.adk7140. Table S3.

Reviewer #2 (Remarks to the Author):

The authors have comprehensively addressed all of my comments. I recommend publication in the current form.

Response: Thank you for your support and positive comments.

Reviewer #3 (Remarks to the Author):

The authors improved the interpretation of their results and highlighted more clearly that what they observe is of physiological relevance. I have no more objections to publish the improved version of this article.

Response: Thank you for your support and valuable comments.